# Achievable Fairness on Your Data With Utility Guarantees

**Muhammad Faaiz Taufiq**[*]
ByteDance Research
faaiz.taufiq@bytedance.com

**Jean-François Ton**
ByteDance Research
jeanfrancois@bytedance.com

**Yang Liu**
University of California Santa Cruz
yangliu@ucsc.edu

## Abstract

In machine learning fairness, training models that minimize disparity across different sensitive groups often leads to diminished accuracy, a phenomenon known as the fairness-accuracy trade-off. The severity of this trade-off inherently depends on dataset characteristics such as dataset imbalances or biases and therefore, using a uniform fairness requirement across diverse datasets remains questionable. To address this, we present a computationally efficient approach to approximate the fairness-accuracy trade-off curve tailored to individual datasets, backed by rigorous statistical guarantees. By utilizing the You-Only-Train-Once (YOTO) framework, our approach mitigates the computational burden of having to train multiple models when approximating the trade-off curve. Crucially, we introduce a novel methodology for quantifying uncertainty in our estimates, thereby providing practitioners with a robust framework for auditing model fairness while avoiding false conclusions due to estimation errors. Our experiments spanning tabular (e.g., `Adult`), image (`CelebA`), and language (`Jigsaw`) datasets underscore that our approach not only reliably quantifies the optimum achievable trade-offs across various data modalities but also helps detect suboptimality in SOTA fairness methods.

## 1 Introduction

A key challenge in fairness for machine learning is to train models that minimize disparity across various sensitive groups such as race or gender [9, 35, 10]. This often comes at the cost of reduced model accuracy, a phenomenon termed accuracy-fairness trade-off [36, 32]. This trade-off can differ significantly across datasets, depending on factors such as dataset biases, imbalances etc. [1, 8, 11].

To demonstrate how these trade-offs are inherently dataset-dependent, we consider a simple example involving two distinct crime datasets. Dataset A has records from a community where crime rates are uniformly distributed across all racial groups, whereas Dataset B comes from a community where historical factors have resulted in a disproportionate crime rate among a specific racial group. Intuitively, training models which are racially agnostic is more challenging for Dataset B, due to the unequal distribution of crime rates across racial groups, and will result in a greater loss in model accuracy as compared to Dataset A.

This example underscores that setting a uniform fairness requirement across diverse datasets (such as requiring the fairness violation metric to be below 10% for both datasets), while also adhering to essential accuracy benchmarks is impractical. Therefore, choosing fairness guidelines for any

---

[*]Corresponding authors: faaiz.taufiq@bytedance.com and jeanfrancois@bytedance.com

38th Conference on Neural Information Processing Systems (NeurIPS 2024).

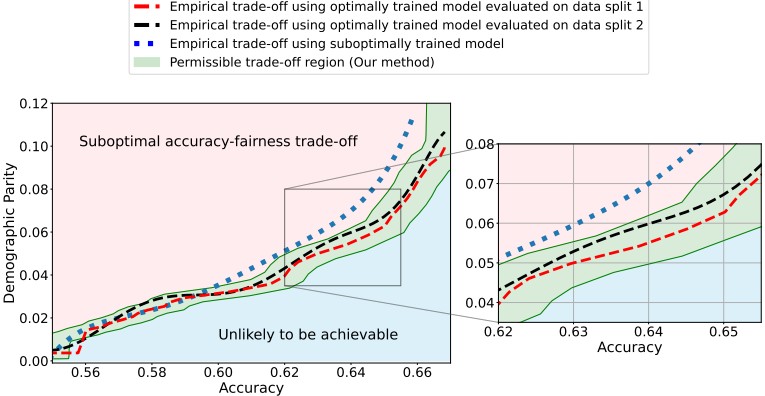

Figure 1: Accuracy-fairness trade-offs for COMPAS dataset (on held-out data). The black and red curves are obtained using the same optimally trained model evaluated on different splits. The blue curve is obtained using a suboptimally trained model. The green area depicts the range of permissible fairness violations for each accuracy, pink area shows suboptimal accuracy-fairness trade-offs, and blue area shows unlikely-to-be-achieved ones. (Details in Appendix F.5)

dataset necessitates careful consideration of its individual characteristics and underlying biases. In this work, we advocate against the use of one-size-fits-all fairness mandates by proposing a nuanced, dataset-specific framework for quantifying acceptable range of accuracy-fairness trade-offs. To put it concretely, the question we consider is:

*Given a dataset, what is the range of permissible fairness violations corresponding to each accuracy threshold for models in a given class $\mathcal{H}$?*

This question can be addressed by considering the optimum accuracy-fairness trade-off, which shows the minimum fairness violation achievable for each level of accuracy. Unfortunately, this curve is typically unavailable and hence, various optimization techniques have been proposed to approximate this curve, ranging from regularization [8, 33] to adversarial learning [45, 40].

However, approximating the trade-off curve using these aforementioned methods has some serious limitations. Firstly, these methods require retraining hundreds if not thousands of models to obtain a good approximation of the trade-off curve, making them computationally infeasible for large datasets or models. Secondly, these works do not account for finite-sampling errors in the obtained curve. This is problematic since the empirical trade-off evaluated over a finite dataset may not match the exact trade-off over the full data distribution.

We illustrate this phenomenon in Figure 1 where the black and red trade-off curves are obtained using the same model but evaluated over two different test data draws. Here, relying solely on the estimated curves without accounting for the uncertainty could lead us to make the incorrect conclusion that the methodology used to obtain the black trade-off curve is sub-optimal (compared to the red curve) as it achieves a higher fairness violation for accuracies in the range $[0.62, 0.66]$. However, this discrepancy arises solely due to finite-sampling errors.

In this paper, we address these challenges by introducing a computationally efficient method of approximating the optimal accuracy-fairness trade-off curve, supported by rigorous statistical guarantees. Our methodology not only circumvents the need to train multiple models (leading to at least a 10-fold reduction in computational cost) but is also the first to quantify the uncertainty in the estimated curve, arising from both, finite-sampling error as well as estimation error. To achieve this, our approach adopts a novel probabilistic perspective and provides guarantees that remain valid across all finite-sample draws. This also allows practitioners to distinguish if an apparent suboptimality in a baseline could be explained by finite-sampling errors (as in the black curve in Figure 1), or if it stems from genuine deficiencies in the fairness interventions applied (as in the blue curve).

The contributions of this paper are three-fold:

- We present a computationally efficient methodology for approximating the accuracy-fairness trade-off curve by training only a single model. This is achieved by adapting a technique from [15] called You-Only-Train-Once (YOTO) to the fairness setting.

- To account for the approximation and finite-sampling errors, we introduce a novel technical framework to construct confidence intervals (using the trained YOTO model) which contain the optimal accuracy-fairness trade-off curve with statistical guarantees. For any accuracy threshold $\psi$ chosen at *inference time*, this gives us a statistically backed range of permissible fairness violations $[l(\psi), u(\psi)]$, allowing us to answer our previously posed question:

  *Given a dataset, the permissible range of fairness violations corresponding to an accuracy threshold of $\psi$ is $[l(\psi), u(\psi)]$ for models in a given class $\mathcal{H}$.*

- Lastly, we showcase the vast applicability of our method empirically across various data modalities including tabular, image and text datasets. We evaluate our framework on a suite of SOTA fairness methods and show that our intervals are both reliable and informative.

## 2 Preliminaries

**Notation** Throughout this paper, we consider a binary classification task, where each training sample is composed of triples, $(X, A, Y)$. $X \in \mathcal{X}$ denotes a vector of features, $A \in \mathcal{A}$ indicates a discrete sensitive attribute, and $Y \in \mathcal{Y} := \{0, 1\}$ represents a label. To make this more concrete, if we take loan default prediction as the classification task, $X$ represents individuals' features such as their income level and loan amount; $A$ represents their racial identity; and $Y$ represents their loan default status. Having established the notation, for completeness, we provide some commonly used fairness violations $\Phi_{\text{fair}}(h) \in [0, 1]$ for a classifier model $h : \mathcal{X} \to \mathcal{Y}$ when $\mathcal{A} = \{0, 1\}$:

**Demographic Parity (DP)** The DP condition states that the selection rates for all sensitive groups are equal, i.e. $\mathbb{P}(h(X) = 1 \mid A = a) = \mathbb{P}(h(X) = 1)$ for any $a \in \mathcal{A}$. The absolute DP violation is:

$$\Phi_{\text{DP}}(h) := |\mathbb{P}(h(X) = 1 \mid A = 1) - \mathbb{P}(h(X) = 1 \mid A = 0)|.$$

**Equalized Opportunity (EOP)** The EOP condition states that the true positive rates for all sensitive groups are equal, i.e. $\mathbb{P}(h(X) = 1 \mid A = a, Y = 1) = \mathbb{P}(h(X) = 1 \mid Y = 1)$ for any $a \in \mathcal{A}$. The absolute EOP violation is:

$$\Phi_{\text{EOP}}(h) := |\mathbb{P}(h(X) = 1 \mid A = 1, Y = 1) - \mathbb{P}(h(X) = 1 \mid A = 0, Y = 1)|.$$

### 2.1 Problem setup

Next, we formalise the notion of *accuracy-fairness trade-off*, which is the main quantity of interest in our work. For a model class $\mathcal{H}$ (e.g., neural networks) and a given accuracy threshold $\psi \in [0, 1]$, we define the optimal accuracy-fairness trade-off $\tau^*_{\text{fair}}(\psi)$ as,

$$\tau^*_{\text{fair}}(\psi) := \min_{h \in \mathcal{H}} \Phi_{\text{fair}}(h) \quad \text{subject to} \quad \text{acc}(h) \geq \psi. \tag{1}$$

Here, $\Phi_{\text{fair}}(h)$ and $\text{acc}(h)$ denote the fairness violation and accuracy of $h$ over the *full data distribution*. For an accuracy $\psi'$ which is unattainable, we define $\tau^*_{\text{fair}}(\psi') = 1$ and we focus on models $\mathcal{H}$ trained using gradient-based methods. Crucially, our goal is not to estimate the trade-off at a fixed accuracy level, but instead to reliably and efficiently estimate the *entire* trade-off curve $\tau^*_{\text{fair}}$. In contrast, previous works [1, 11] impose an apriori fairness constraint during training and therefore each trained model only recovers one point on the trade-off curve corresponding to this pre-specified constraint.

If available, this trade-off curve would allow practitioners to characterise exactly how, for a given dataset, the minimum fairness violation varies as model accuracy increases. This not only provides a principled way of selecting data-specific fairness requirements, but also serves as a tool to audit if a model meets acceptable fairness standards by checking if its accuracy-fairness trade-off lies on this curve. Nevertheless, obtaining this ground-truth trade-off curve exactly is impossible within the confines of a finite-sample regime (owing to finite-sampling errors). This means that even if baseline A's empirical trade-off evaluated on a finite dataset is suboptimal compared to the empirical trade-off of baseline B, this does not necessarily imply suboptimality on the full data distribution.

We illustrate this in Figure 1 where both red and black trade-off curves are obtained using the same model but evaluated on different test data splits. Here, even though the black curve appears suboptimal

compared to the red curve for accuracies in $[0.62, 0.66]$, this apparent suboptimality is solely due to finite-sampling errors (since the discrepancy between the two curves arises only due to different evaluation datasets). If we rely only on comparing empirical trade-offs, we would incorrectly flag the methodology used to obtain the black curve as suboptimal.

To address this, we construct confidence intervals (CIs), shown as the green region in Figure 1, that account for such finite-sampling errors. In this case both trade-offs fall within our CIs which correctly indicates that this apparent suboptimality could stem from finite-sample variability. Conversely, a baseline's trade-off falling above our CIs (as in the blue curve in Figure 1) offers a confident assessment of suboptimality, as this cannot be explained away by finite-sample variability. Therefore, our CIs equip practitioners with a robust auditing tool. They can confidently identify suboptimal baselines while avoiding false conclusions caused by considering empirical trade-offs alone.

**High-level road map** To achieve this, our proposed methodology adopts a two-step approach:

1. Firstly, we propose *loss-conditional fairness training*, a computationally efficient methodology of estimating the entire trade-off curve $\tau_{\text{fair}}^*$ by training a single model, obtained by adapting the YOTO framework [15] to the fairness setting.

2. Secondly, to account for the approximation and finite-sampling errors in our estimates, we introduce a novel methodology of constructing confidence intervals on the trade-off curve $\tau_{\text{fair}}^*$ using the trained YOTO model. Specifically, given $\alpha \in (0, 1)$, we construct confidence intervals $\Gamma_{\text{fair}}^\alpha \subseteq [0, 1]$ which satisfy guarantees of the form:

$$\mathbb{P}(\tau_{\text{fair}}^*(\Psi) \in \Gamma_{\text{fair}}^\alpha) \geq 1 - \alpha.$$

   Here, $\Gamma_{\text{fair}}^\alpha$ and $\Psi \in [0, 1]$ are random variables obtained using a held-out calibration dataset $\mathcal{D}_{\text{cal}}$ (see Section 3.2) and the probability is taken over different draws of $\mathcal{D}_{\text{cal}}$.

## 3 Methodology

First, we demonstrate how our 2-step approach offers a practical and statistically sound method for estimating $\tau_{\text{fair}}^*(\psi)$. Figure 1 provides an illustration of our proposed confidence intervals (CIs) $\Gamma_{\text{fair}}^\alpha$ and shows how they can be interpreted as a range of 'permissible' values of accuracy-fairness trade-offs (the green region). Specifically, if for a classifier $h_0$, the accuracy-fairness pair $(\text{acc}(h_0), \Phi_{\text{fair}}(h_0))$ lies above the CIs $\Gamma_{\text{fair}}^\alpha$ (i.e., the pink region in Figure 1), then $h_0$ is likely to be suboptimal in terms of the fairness violation, i.e., there likely exists $h' \in \mathcal{H}$ with $\text{acc}(h') \geq \text{acc}(h_0)$ and $\Phi_{\text{fair}}(h') \leq \Phi_{\text{fair}}(h_0)$. On the other hand, it is unlikely for any model $h' \in \mathcal{H}$ to achieve a trade-off below the CIs $\Gamma_{\text{fair}}^\alpha$ (the blue region in Figure 1). Next, we outline how to construct such intervals.

### 3.1 Step 1: Efficient estimation of trade-off curve

The first step of constructing the intervals is to approximate the trade-off curve by recasting the problem into a constrained optimization objective. The optimization problem formulated in Eq. (1) is however, often too complex to solve, because the accuracy $\text{acc}(h)$ and fairness violations $\Phi_{\text{fair}}(h)$ are both non-smooth [1]. These constraints make it hard to use standard optimization methods that rely on gradients [25]. To get around this issue, previous works [1, 8] replace the non-smooth constrained optimisation problem with a smooth surrogate loss. Here, we consider parameterized family of classifiers $\mathcal{H} = \{h_\theta : \mathcal{X} \to \mathbb{R} \,|\, \theta \in \Theta\}$ (such as neural networks) trained using the regularized loss:

$$\mathcal{L}_\lambda(\theta) = \mathbb{E}[l_{\text{CE}}(h_\theta(X), Y)] + \lambda \mathcal{L}_{\text{fair}}(h_\theta). \tag{2}$$

where, $l_{\text{CE}}$ is the cross-entropy loss for the classifier $h_\theta$ and $\mathcal{L}_{\text{fair}}(h_\theta)$ is a smooth relaxation of the fairness violation $\Phi_{\text{fair}}$ [8, 29]. For example, when the fairness violation is DP, [8] consider

$$\mathcal{L}_{\text{fair}}(h_\theta) = \mathbb{E}[g(h_\theta(X)) \mid A = 1] - \mathbb{E}[g(h_\theta(X)) \mid A = 0],$$

for different choices of $g(x)$, including the identity and sigmoid functions. We include more examples of such regularizers in Appendix F.3. The parameter $\lambda$ in $\mathcal{L}_\lambda$ modulates the accuracy-fairness trade-off with lower values of $\lambda$ favouring higher accuracy over reduced fairness violation.

Now that we defined the optimization objective, obtaining the trade-off curve becomes straightforward by simply optimizing multiple models over a grid of regularization parameters $\lambda$. However, training multiple models can be computationally expensive, especially when this involves large-scale models (e.g. neural networks). To circumvent this computational challenge, we introduce loss-conditional fairness training obtained by adapting the YOTO framework proposed by [15].

### 3.1.1 Loss-conditional fairness training

As we describe above, a popular approach for approximating the accuracy-fairness trade-off $\tau^*_{\text{fair}}(\psi)$ involves training multiple models $h_{\theta^*_\lambda}$ over a discrete grid of $\lambda$ hyperparameters with the regularized loss $\mathcal{L}_\lambda$. To avoid the computational overhead of training multiple models, [15] propose 'You Only Train Once' (YOTO), a methodology of training one model $h_\theta : \mathcal{X} \times \Lambda \to \mathbb{R}$, which takes $\lambda \in \Lambda \subseteq \mathbb{R}$ as an additional input using Feature-wise Linear Modulation (FiLM) [34] layers. YOTO is trained such that at inference time $h_\theta(\cdot, \lambda')$ recovers the classifier obtained by minimising $\mathcal{L}_{\lambda'}$ in Eq. (2).

Recall that we are interested in minimising the family of losses $\mathcal{L}_\lambda$, parameterized by $\lambda \in \Lambda$ (Eq. (2)). Instead of fixing $\lambda$, YOTO solves an optimisation problem where the parameter $\lambda$ is sampled from a distribution $P_\lambda$. As a result, during training the model observes many values of $\lambda$ and learns to optimise the loss $\mathcal{L}_\lambda$ for all of them simultaneously. At inference time, the model can be conditioned on a chosen value $\lambda'$ and recovers the model trained to optimise $\mathcal{L}_{\lambda'}$. Hence, once adapted to our setting, the YOTO loss becomes:

$$\underset{h_\theta : \mathcal{X} \times \Lambda \to \mathbb{R}}{\arg\min} \; \mathbb{E}_{\lambda \sim P_\lambda} \left[ \mathbb{E}[l_{\text{CE}}(h_\theta(X, \lambda), Y)] + \lambda \mathcal{L}_{\text{fair}}(h_\theta(\cdot, \lambda)) \right].$$

Having trained a YOTO model, the trade-off curve $\tau^*_{\text{fair}}(\psi)$ can be approximated by simply plugging in different values of $\lambda$ at inference time and thus avoiding additional training. From a theoretical point of view, [15, Proposition 1] proves that under the assumption of large enough model capacity, training the loss-conditional YOTO model performs as well as the separately trained models while only requiring a single model. Although the model capacity assumption might be hard to verify in practice, our experimental section has shown that the trade-off curves estimates $\widehat{\tau^*_{\text{fair}}(\psi)}$ obtained using YOTO are consistent with the ones obtained using separately trained models.

It should be noted, as is common in optimization problems, that the estimated trade-off curve $\widehat{\tau^*_{\text{fair}}(\psi)}$ may not align precisely with the true trade-off curve $\tau^*_{\text{fair}}(\psi)$. This discrepancy originates from two key factors. Firstly, the limited size of the training and evaluation datasets introduces errors in the estimation of $\widehat{\tau^*_{\text{fair}}(\psi)}$. Secondly, we opt for a computationally tractable loss function instead of the original optimization problem in Eq. (1). This may result in our estimation $\widehat{\tau^*_{\text{fair}}(\psi)}$ yielding sub-optimal trade-offs, as can be seen from Figure 1. Therefore, to ensure that our procedure yields statistically sound inferences, we next construct confidence intervals using the YOTO model, designed to contain the true trade-off curve $\tau^*_{\text{fair}}(\psi)$ with high probability.

### 3.2 Step 2: Constructing confidence intervals

As mentioned above, our goal here is to use our trained YOTO model to construct confidence intervals (CIs) for the optimal trade-off curve $\tau^*_{\text{fair}}(\psi)$ defined in Eq. (1). Specifically, we assume access to a held-out *calibration* dataset $\mathcal{D}_{\text{cal}} := \{(X_i, A_i, Y_i)\}_i$ which is disjoint from the training data. Given a level $\alpha \in [0, 1]$, we construct CIs $\Gamma^\alpha_{\text{fair}} \subseteq [0, 1]$ using $\mathcal{D}_{\text{cal}}$, which provide guarantees of the form:

$$\mathbb{P}(\tau^*_{\text{fair}}(\Psi) \in \Gamma^\alpha_{\text{fair}}) \geq 1 - \alpha. \tag{3}$$

Here, it is important to note that $\Psi \in [0, 1]$ and $\Gamma^\alpha_{\text{fair}}$ are random variables obtained from the calibration data $\mathcal{D}_{\text{cal}}$, and the guarantee in Eq. (3) holds marginally over $\Psi$ and $\Gamma^\alpha_{\text{fair}}$. While our CIs in this section require the availability of the sensitive attributes in $\mathcal{D}_{\text{cal}}$, in Appendix D we also extend our methodology to the setting where sensitive attributes are missing. In this section, for notational convenience we use $h_\lambda(\cdot)$ to denote the YOTO model $h_\theta(\cdot, \lambda)$ for $\lambda \in \Lambda$.

The uncertainty in our trade-off estimate arises, in part, from the uncertainty in the accuracy and fairness violations of our trained model. Therefore, our methodology of constructing CIs on $\tau^*_{\text{fair}}$, involves first constructing CIs on test accuracy $\text{acc}(h_\lambda)$ and fairness violation $\Phi_{\text{fair}}(h_\lambda)$ for a given value of $\lambda$ using $\mathcal{D}_{\text{cal}}$, denoted as $C^\alpha_{\text{acc}}(\lambda)$ and $C^\alpha_{\text{fair}}(\lambda)$ respectively satisfying,

$$\mathbb{P}(\text{acc}(h_\lambda) \in C^\alpha_{\text{acc}}(\lambda)) \geq 1 - \alpha, \quad \text{and} \quad \mathbb{P}(\Phi_{\text{fair}}(h_\lambda) \in C^\alpha_{\text{fair}}(\lambda)) \geq 1 - \alpha.$$

One way to construct these CIs involves using assumption-light concentration inequalities such as Hoeffding's inequality. To be more concrete, for the accuracy $\text{acc}(h_\lambda)$:

**Lemma 3.1** (Hoeffding's inequality). *Given a classifier $h_\lambda : \mathcal{X} \to \mathcal{Y}$, we have that,*

$$\mathbb{P}\left(\text{acc}(h_\lambda) \in \left[\widetilde{\text{acc}(h_\lambda)} - \delta, \widetilde{\text{acc}(h_\lambda)} + \delta\right]\right) \geq 1 - \alpha.$$

*Here, $\widetilde{\text{acc}(h)} := \sum_{(X_i, A_i, Y_i) \in \mathcal{D}_{\text{cal}}} \frac{\mathbb{1}(h(X_i)=Y_i)}{|\mathcal{D}_{\text{cal}}|}$ and $\delta := \sqrt{\frac{1}{2|\mathcal{D}_{\text{cal}}|} \log\left(\frac{2}{\alpha}\right)}$.*

Lemma 3.1 illustrates that we can use Hoeffding's inequality to construct confidence interval $C_{\text{acc}}^{\alpha}(\lambda) = [\widetilde{\text{acc}(h_\lambda)} - \delta, \widetilde{\text{acc}(h_\lambda)} + \delta]$ on $\text{acc}(h_\lambda)$ such that the true $\text{acc}(h_\lambda)$ will lie inside the CI with probability $1 - \alpha$. Analogously, we also construct CIs for fairness violations, $\Phi_{\text{fair}}(h_\lambda)$, although this is subject to additional nuanced challenges, which we address using a novel sub-sampling based methodology in Appendix B. Once we have CIs over $\text{acc}(h_\lambda)$ and $\Phi_{\text{fair}}(h_\lambda)$ for a model $h_\lambda$, we next outline how to use these to derive CIs for the minimum achievable fairness $\tau_{\text{fair}}^*$, satisfying Eq. (3). We proceed by explaining how to construct the upper and lower CIs separately, as the latter requires additional considerations regarding the trade-off achieved by YOTO.

### 3.2.1 Upper confidence intervals

We first outline how to obtain one-sided upper confidence intervals on the optimum accuracy-fairness trade-off $\tau_{\text{fair}}^*(\Psi)$ of the form $\Gamma_{\text{fair}}^{\alpha} = [0, U_{\text{fair}}^{\lambda}]$, which satisfies the probabilistic guarantee in Eq. (3). To this end, given a classifier $h_\lambda \in \mathcal{H}$, our methodology involves constructing one-sided lower CI on the accuracy $\text{acc}(h_\lambda)$ and upper CI on the fairness violation $\Phi_{\text{fair}}(h_\lambda)$. We make this concrete below:

**Proposition 3.2.** *Given $h_\lambda \in \mathcal{H}$, let $L_{\text{acc}}^{\lambda}, U_{\text{fair}}^{\lambda} \in [0, 1]$ be lower and upper CIs on $\text{acc}(h_\lambda)$ and $\Phi_{\text{fair}}(h_\lambda)$, i.e.*

$$\mathbb{P}\left(\text{acc}(h_\lambda) \geq L_{\text{acc}}^{\lambda}\right) \geq 1 - \alpha/2, \quad \text{and} \quad \mathbb{P}(\Phi_{\text{fair}}(h_\lambda) \leq U_{\text{fair}}^{\lambda}) \geq 1 - \alpha/2.$$

*Then, $\mathbb{P}\left(\tau_{\text{fair}}^*(L_{\text{acc}}^{\lambda}) \leq U_{\text{fair}}^{\lambda}\right) \geq 1 - \alpha$.*

Proposition 3.2 shows that for any model $h_\lambda$, the upper CI on model fairness, $U_{\text{fair}}^{\lambda}$, provides a valid upper CI for the trade-off value at $L_{\text{acc}}^{\lambda}$, i.e. $\tau_{\text{fair}}^*(L_{\text{acc}}^{\lambda})$. This can be used to construct upper CIs on $\tau_{\text{fair}}^*(\psi)$ for a given accuracy level $\psi$. To understand how this can be achieved, we first find $\lambda \in \Lambda$ such that the lower CI on the accuracy of model $h_\lambda$, $L_{\text{acc}}^{\lambda}$, satisfies $L_{\text{acc}}^{\lambda} \geq \psi$. Then, since by definition $\tau_{\text{fair}}^*$ is a monotonically increasing function, we know that $\tau_{\text{fair}}^*(L_{\text{acc}}^{\lambda}) \geq \tau_{\text{fair}}^*(\psi)$. Since Proposition 3.2 tells us that $U_{\text{fair}}^{\lambda}$ is an upper CI for $\tau_{\text{fair}}^*(L_{\text{acc}}^{\lambda})$, it follows that $U_{\text{fair}}^{\lambda}$ is also a valid upper CI for $\tau_{\text{fair}}^*(\psi)$.

Intuitively, Proposition 3.2 provides the 'worst-case' optimal trade-off, accounting for finite-sample uncertainty. It is important to note that this result does not rely on any assumptions regarding the optimality of the trained classifiers. This means that the upper CIs will remain valid even if the YOTO classifier $h_\lambda$ is not trained well (and hence achieves sub-optimal accuracy-fairness trade-offs), although in such cases the CI may be conservative.

Having explained how to construct upper CIs on $\tau_{\text{fair}}^*(\psi)$, we next move on to the lower CIs.

### 3.2.2 Lower confidence intervals

Obtaining lower confidence intervals on $\tau_{\text{fair}}^*(\psi)$ is more challenging than obtaining upper confidence intervals. We begin by explaining at an intuitive level why this is the case.

Suppose that $h_\lambda \in \mathcal{H}$ is such that $\text{acc}(h_\lambda) = \psi$, then since $\tau_{\text{fair}}^*$ denotes the minimum attainable fairness violation (Eq. (1)), we have that $\tau_{\text{fair}}^*(\psi) \leq \Phi_{\text{fair}}(h_\lambda)$. Therefore, any valid upper confidence interval on $\Phi_{\text{fair}}(h_\lambda)$ will also be valid for $\tau_{\text{fair}}^*(\psi)$. However, a lower bound on $\Phi_{\text{fair}}(h_\lambda)$ cannot be used as a lower bound for the minimum achievable fairness $\tau_{\text{fair}}^*(\psi)$ in general. A valid lower CI for $\tau_{\text{fair}}^*(\psi)$ will therefore depend on the gap between fairness violation achieved by $h_\lambda$, $\Phi_{\text{fair}}(h_\lambda)$, and minimum achievable fairness violation $\tau_{\text{fair}}^*(\psi)$ (i.e., $\Delta(h_\lambda)$ term in Figure 2a). We make this concrete by constructing lower CIs depending on $\Delta(h_\lambda)$ explicitly.

**Proposition 3.3.** *Given $h_\lambda \in \mathcal{H}$, let $U_{\text{acc}}^{\lambda}, L_{\text{fair}}^{\lambda} \in [0, 1]$ be upper and lower CIs on $\text{acc}(h_\lambda)$ and $\Phi_{\text{fair}}(h_\lambda)$, i.e.*

$$\mathbb{P}(\text{acc}(h_\lambda) \leq U_{\text{acc}}^{\lambda}) \geq 1 - \alpha/2, \quad \text{and} \quad \mathbb{P}(\Phi_{\text{fair}}(h_\lambda) \geq L_{\text{fair}}^{\lambda}) \geq 1 - \alpha/2.$$

*Then, $\mathbb{P}\left(\tau_{\text{fair}}^*(U_{\text{acc}}^{\lambda}) \geq L_{\text{fair}}^{\lambda} - \Delta(h_\lambda)\right) \geq 1 - \alpha$, where $\Delta(h_\lambda) := \Phi_{\text{fair}}(h_\lambda) - \tau_{\text{fair}}^*(\text{acc}(h_\lambda)) \geq 0$.*

Proposition 3.3 can be used to derive lower CIs on $\tau_{\text{fair}}^*(\psi)$ at a specified accuracy level $\psi$, using a methodology analogous to that described in Section 3.2.1. Intuitively, this result provides the 'best-case' optimal trade-off, accounting for finite-sample uncertainty. However, unlike the upper CI, the lower CI includes the $\Delta(h_\lambda)$ term, which is typically unknown. To circumvent this, we propose a strategy for obtaining plausible approximations for $\Delta(h_\lambda)$ in practice in the following section.

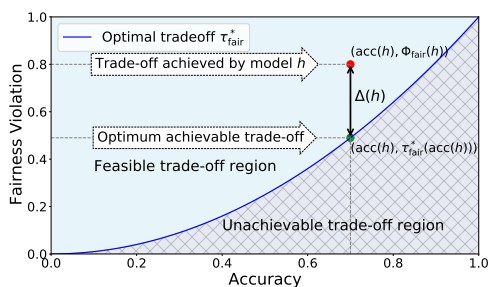 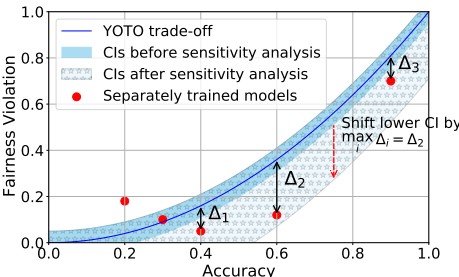

(a) Visual representation of the difference between the optimal and model achieved trade-offs.

(b) Sensitivity analysis shifts lower CIs below by a constant amount whereas upper CIs remain unchanged.

Figure 2: Visual illustrations for $\Delta(h)$ (Figure 2a) and our sensitivity analysis procedure (Figure 2b).

### 3.2.3 Sensitivity analysis for $\Delta(h_\lambda)$

Recall that $\Delta(h_\lambda)$ quantifies the difference between the fairness loss of classifier $h_\lambda$ and the minimum attainable fairness loss $\tau^*_{\text{fair}}(\text{acc}(h_\lambda))$, and is an unknown quantity in general (see Figure 2a). Here, we propose a practical strategy for positing values for $\Delta(h_\lambda)$ which encode our belief on how close the fairness loss $\Phi_{\text{fair}}(h_\lambda)$ is to $\tau^*_{\text{fair}}(\text{acc}(h_\lambda))$. This allows us to construct CIs which not only incorporate finite-sampling uncertainty from calibration data, but also account for the possible sub-optimality in the trade-offs achieved by $h_\lambda$. The main idea behind our approach is to calibrate $\Delta(h_\lambda)$ using additional separately trained standard models without imposing significant computational overhead.

**Details**   Our sensitivity analysis uses $k$ additional models $\mathcal{M} := \{h^{(1)}, h^{(2)}, \ldots, h^{(k)}\} \subseteq \mathcal{H}$ trained separately using the standard regularized loss $\mathcal{L}_{\lambda'}$ (Eq. (2)) for some randomly chosen values of $\lambda'$. Let $\mathcal{M}_0 \subseteq \mathcal{M}$ denote the models which achieve a better empirical trade-off than the YOTO model on $\mathcal{D}_{\text{cal}}$, i.e. the empirical trade-offs for models in $\mathcal{M}_0$ lie below the YOTO trade-off curve (see Figure 2b). We choose $\Delta(h_\lambda)$ for our YOTO model to be the maximum gap between empirical trade-offs of these separately trained models in $\mathcal{M}_0$ and the YOTO model. It can be seen from Proposition 3.3 that, in practice, this will result in a downward shift in the lower CI until all the separately trained models in $\mathcal{M}$ lie above the lower CI. As a result, our methodology yields increasingly conservative lower CIs as the number of additional models $|\mathcal{M}|$ increases.

Even though the procedure above requires training additional models $\mathcal{M}$, it does not impose the same computational overhead as training models over the full range of $\lambda$ values. We show empirically in Section 5 that in practice 2 models are usually sufficient to obtain informative and reliable intervals. Additionally, we also show that when YOTO achieves the optimal trade-off (i.e., $\Delta(h_\lambda) = 0$), our sensitivity analysis leaves the CIs unchanged, thereby preventing unnecessary conservatism.

**Asymptotic analysis of $\Delta(h_\lambda)$**   While the procedure described above provides a practical solution for obtaining plausible approximations for $\Delta(h_\lambda)$, we next present a theoretical result which provides reassurance that this gap should become negligible as the number of training data $\mathcal{D}_{\text{tr}}$ increases.

**Theorem 3.4.** *Let* $\widehat{\Phi_{\text{fair}}(h')}, \widehat{\text{acc}(h')}$ *denote the fairness violation and accuracy for* $h' \in \mathcal{H}$ *evaluated on training data* $\mathcal{D}_{\text{tr}}$ *and let*

$$h := \arg \min_{h' \in \mathcal{H}} \widehat{\Phi_{\text{fair}}(h')} \text{ subject to } \widehat{\text{acc}(h')} \geq \delta. \tag{4}$$

*Then, given* $\eta \in (0, 1)$*, under standard regularity assumptions, we have that with probability at least* $1 - \eta$*,* $\Delta(h) \leq \widetilde{O}(|\mathcal{D}_{\text{tr}}|^{-\gamma})$*, for some* $\gamma \in (0, 1/2]$ *where* $\widetilde{O}(\cdot)$ *suppresses dependence on* $\log(1/\eta)$*.*

Theorem 3.4 shows that as the training data size $|\mathcal{D}_{\text{tr}}|$ increases, the error term $\Delta(h_\lambda)$ will become negligible with a high probability for any model $h_\lambda$ which minimises the empirical training loss in Eq. (4). In this case, $\Delta(h_\lambda)$ should not have a significant impact on the lower CIs in Proposition 3.3 and the CIs will reflect the uncertainty in $\tau^*_{\text{fair}}$ arising mostly due to finite calibration data. We also verify this empirically in Appendix F.7. It is worth noting that Theorem 3.4 relies on the same assumptions as used in Theorem 2 in [1], which have been provided in Appendix A.2.

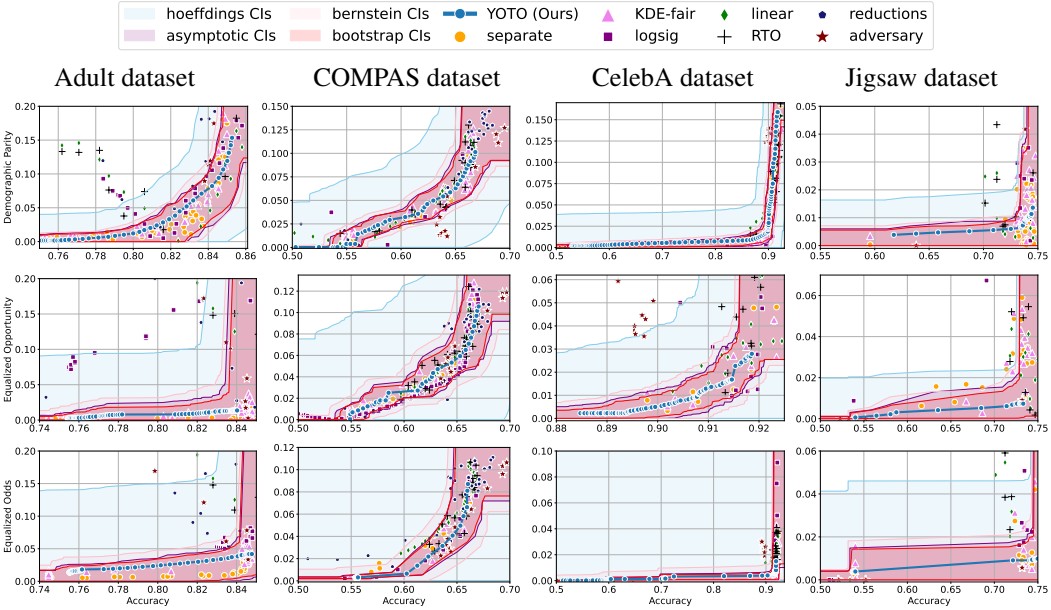

Figure 3: Results on four real-world datasets where $\mathcal{D}_{\text{cal}}$ is a 10% data split. Here, $\alpha = 0.05$ and we use $|\mathcal{M}| = 2$ separately trained models for sensitivity analysis.

# 4 Related works

Many previous fairness methods in the literature, termed in-processing methods, introduce constraints or regularization terms to the optimization objective. For instance, [1, 10] impose a priori uniform constraints on model fairness at training time. However, given the data-dependent nature of accuracy-fairness trade-offs, setting a uniform fairness threshold may not be suitable. Other in-processing methods [37, 24] consider information-theoretic bounds on the optimal trade-off in infinite data limit, independent of a specific model class. While these works offer valuable theoretical insights, there is no guarantee that these frontiers are attainable by models within a given model class $\mathcal{H}$. We verify this empirically in Appendix F.6 by showing that, for the Adult dataset, the frontiers proposed in [24] are not achieved by any SOTA method we considered. In contrast, our method provides guarantees on the *achievable* trade-off curve within realistic constraints of model class and data availability.

Various other regularization approaches [39, 33, 8, 14, 41, 42, 43] have also been proposed, but these often necessitate training multiple models, making them computationally intensive. Alternative strategies include learning 'fair' representations [44, 30, 31], or re-weighting data based on sensitive attributes [18, 22]. These, however, provide limited control over accuracy-fairness trade-offs.

Besides this, post-processing methods [19, 38] enforce fairness after training but can lead to other forms of unfairness such as disparate treatment of similar individuals [16]. Moreover, many post-hoc approaches such as [3, 2] still require solving different optimisation problems for different fairness thresholds. Other methods such as [46, 27] involve learning a post-hoc module in addition to the base classifier. As a result, the computational cost of training the YOTO model is similar to (and in many cases lower than) the combined cost of training a base model and subsequently applying a post-processing intervention to this pre-trained classifier. We confirm this empirically in Section 5.

# 5 Experiments

In this section, we empirically validate our methodology of constructing confidence intervals on the fairness trade-off curve, across diverse datasets with neural networks as model class $\mathcal{H}$. These datasets range from tabular (`Adult` and `COMPAS` ), to image-based (`CelebA`), and natural language processing datasets (`Jigsaw`). Recall that our approach involves two steps: initial estimation of the trade-off via the YOTO model, followed by the construction of CIs using calibration data $\mathcal{D}_{\text{cal}}$.

Table 1: Proportion of empirical trade-offs for each baseline in the three trade-off regions, aggregated across all datasets and fairness metrics (using Bernstein's CIs). 'Unlikely', 'Permissible' and 'Sub-optimal' correspond to the blue, green and pink regions in Figure 1 respectively. The last column shows the rough average training time per model across experiments × no. of models per experiment.

| Category | Baseline | Unlikely | Permissible | Sub-optimal | $\approx$ Training time |
|---|---|---|---|---|---|
| In-processing | adversary [45] | $0.03 \pm 0.03$ | $0.51 \pm 0.07$ | $0.45 \pm 0.07$ | 100 min×40 |
| | logsig [8] | $0.0 \pm 0.0$ | $0.66 \pm 0.1$ | $0.33 \pm 0.1$ | 100 min×40 |
| | reductions [1] | $0.0 \pm 0.0$ | $0.79 \pm 0.1$ | $0.21 \pm 0.1$ | 90 min×40 |
| | linear [8] | $0.01 \pm 0.0$ | $0.85 \pm 0.05$ | $0.14 \pm 0.06$ | 100 min×40 |
| | KDE-fair [12] | $0.0 \pm 0.0$ | $0.97 \pm 0.05$ | $0.03 \pm 0.06$ | 85 min×40 |
| | separate | $0.0 \pm 0.0$ | $0.98 \pm 0.01$ | $0.02 \pm 0.02$ | 100 min×40 |
| | YOTO (Ours) | $0.0 \pm 0.0$ | $1.0 \pm 0.0$ | $0.0 \pm 0.0$ | **105 min×1** |
| Post-processing | RTO [2] | $0.0 \pm 0.0$ | $0.65 \pm 0.2$ | $0.35 \pm 0.05$ | 95 min (training base classifier) + 10min×40 (post-hoc optimisations) |

To evaluate our methodology, we implement a suite of baseline algorithms including SOTA in-processing techniques such as regularization-based approaches [8], a SOTA kernel-density based method [12] (denoted as 'KDE-fair'), as well as the reductions method [1]. Additionally, we also compare against adversarial fairness techniques [45] and a post-processing approach (denoted as 'RTO') [2] and consider the three most prominent fairness metrics: Demographic Parity (DP), Equalized Odds (EO), and Equalized Opportunity (EOP). We provide additional details and results in Appendix F, where we also consider a synthetic setup with tractable $\tau_{\text{fair}}^*$. The code to reproduce our experiments is provided at github.com/faaizT/DatasetFairness.

## 5.1 Results

Figure 3 shows the results for different datasets and fairness violations, obtained using a 10% data split as calibration dataset $\mathcal{D}_{\text{cal}}$. For each dataset, we construct 4 CIs that serve as the upper and lower bounds on the optimal accuracy-fairness trade-off curve. These intervals are computed at a 95% confidence level using various methodologies, including 1) Hoeffding's, 2) Bernstein's inequalities which both offer finite sample guarantees as well as, 3) bootstrapping [17], and 4) asymptotic intervals based on the Central Limit Theorem [26] which are valid asymptotically. There are 4 key takeaways:

**Takeaway 1: Trade-off curves are data dependent.** The results in Figure 3 confirm that the accuracy-fairness trade-offs can vary significantly across the datasets. For example, achieving near-perfect fairness (i.e. $\Phi_{\text{fair}}(h) \approx 0$) seems significantly easier for the Jigsaw dataset than the COMPAS dataset, even as the accuracy increases. Likewise, for Adult and COMPAS, the DP increases gradually with increasing accuracy, whereas for CelebA, the increase is sharp once the accuracy increases above 90%. Therefore, using a uniform fairness threshold across datasets [as in 1] may be too restrictive, and our methodology provides more dataset-specific insights about the entire trade-off curve instead.

**Takeaway 2: Our CIs are both reliable and informative.** Recall that, any trade-off which lies above our upper CIs is guaranteed to be sub-optimal with probability $1 - \alpha$, thereby enabling practitioners to effectively distinguish between genuine sub-optimalities and those due to finite-sample errors. Table 1 lists the proportion of sub-optimal empirical trade-offs for each baseline and provides a principled comparison of the baselines. For example, the adversarial, RTO and logsig baselines have a significantly higher proportion of sub-optimal trade-offs than the KDE-fair and separate baselines.

On the other hand, the validity of our lower CIs depends on the optimality of our YOTO model and the lower CIs may be too tight if YOTO is sub-optimal. Therefore, for the lower CIs to be reliable, it must be unlikely for any baseline to achieve a trade-off below the lower CIs. Table 1 confirms this empirically, as the proportion of models which lie below the lower CIs is negligible. In Appendix E, we also account for the uncertainty in baseline trade-offs when assessing the optimality, hence yielding more robust inferences. The results remain similar to those in Table 1.

**Takeaway 3: YOTO trade-offs are consistent with SOTA.** We observe that the YOTO trade-offs align well with most of the SOTA baselines considered while reducing the computational cost by approximately 40-fold (see the final column of Table 1). In some cases, YOTO even achieves a better trade-off than the baselines considered. See, e.g., the Jigsaw dataset results (especially for EOP). Moreover, we observe that the baselines yield empirical trade-offs which have a high variance as

accuracy increases (see Jigsaw results in Figure 3, for example). This behaviour starkly contrasts the smooth variations exhibited by our YOTO-generated trade-off curves along the accuracy axis.

**Takeaway 4: Sensitivity analysis does not cause unnecessary conservatism.** We use 2 randomly chosen separately trained models to perform our sensitivity analysis for Figure 3. We find that this only causes a shift in lower CIs for 2 out of the 12 trade-off curves presented (i.e. for DP and EO trade-offs on the Adult dataset), leaving the rest of the CIs unchanged. Therefore, in practice sensitivity analysis does not impose significant computational overhead, and only changes the CIs when YOTO achieves a suboptimal trade-off. Additional results have been included in Appendix C.

## 6 Discussion and Limitations

In this work, we propose a computationally efficient approach to capture the accuracy-fairness trade-offs inherent to individual datasets, backed by sound statistical guarantees. Our proposed methodology enables a nuanced and dataset-specific understanding of the accuracy-fairness trade-offs. It does so by obtaining confidence intervals on the accuracy-fairness trade-off, leveraging the computational benefits of the You-Only-Train-Once (YOTO) framework [15]. This empowers practitioners with the ability to, at inference time, specify desired accuracy levels and promptly receive corresponding permissible fairness ranges. By eliminating the need for repetitive model training, we significantly streamline the process of obtaining accuracy-fairness trade-offs tailored to individual datasets.

**Limitations** Despite the evident merits of our approach, it also has some limitations. Firstly, our methodology requires distinct datasets for training and calibration, posing difficulties when data is limited. Under such constraints, the YOTO model might not capture the optimal accuracy-fairness trade-off, and moreover, the resulting confidence intervals could be overly conservative. Secondly, our lower CIs incorporate an unknown term $\Delta(h_\lambda)$. While we propose sensitivity analysis for approximating this term and prove that it is asymptotically negligible under certain mild assumptions in Section 3.2.3, a more exhaustive understanding remains an open question. Exploring informative upper bounds for $\Delta(h_\lambda)$ under weaker conditions is a promising avenue for future investigations.

## Acknowledgments

We would like to express our gratitude to Sahra Ghalebikesabi for her valuable feedback on an earlier draft of this paper. We also thank the anonymous reviewers for their thoughtful and constructive comments, which enhanced the clarity and rigor of our final submission.

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

# A Proofs

## A.1 Confidence intervals on $\tau_{\text{fair}}^*$

*Proof of Lemma 3.1.* This lemma is a straightforward application of Heoffding's inequality. □

*Proof of Proposition 3.2.* Here, we prove the result for general classifiers $h \in \mathcal{H}$. Let $L_{\text{acc}}^h, U_{\text{fair}}^h$ be the lower and upper CIs for $\text{acc}(h)$ and $\Phi_{\text{fair}}(h)$ respectively,

$$\mathbb{P}(\text{acc}(h) \geq L_{\text{acc}}^h) \geq 1 - \alpha/2 \quad \text{and} \quad \mathbb{P}(\Phi_{\text{fair}}(h) \leq U_{\text{fair}}^h) \geq 1 - \alpha/2.$$

Then, using a straightforward application union bounds, we get that

$$\mathbb{P}(\text{acc}(h) \geq L_{\text{acc}}^h, \Phi_{\text{fair}}(h) \leq U_{\text{fair}}^h) \geq 1 - \mathbb{P}(\text{acc}(h) < L_{\text{acc}}^h) - \mathbb{P}(\Phi_{\text{fair}}(h) > U_{\text{fair}}^h)$$
$$\geq 1 - \alpha/2 - \alpha/2 = 1 - \alpha.$$

Using the definition of the optimal fairness-accuracy trade-off $\tau_{\text{fair}}^*$, we get that the event

$$\left\{\text{acc}(h) \geq L_{\text{acc}}^h, \Phi_{\text{fair}}(h) \leq U_{\text{fair}}^h\right\} \quad \text{implies,} \quad \left\{\underbrace{\min\{\Phi_{\text{fair}}(h') \mid h' \in \mathcal{H}, \text{acc}(h') \geq L_{\text{acc}}^h\}}_{\tau_{\text{fair}}^*(L_{\text{acc}}^h)} \leq U_{\text{fair}}^h\right\}.$$

From this, it follows that

$$\mathbb{P}(\tau_{\text{fair}}^*(L_{\text{acc}}^h) \leq U_{\text{fair}}^h) \geq \mathbb{P}(\text{acc}(h) \geq L_{\text{acc}}^h, \Phi_{\text{fair}}(h) \leq U_{\text{fair}}^h) \geq 1 - \alpha.$$

□

*Proof of Proposition 3.3.* We prove the result for general classifiers $h \in \mathcal{H}$. Let $U_{\text{acc}}^h, L_{\text{fair}}^h$ be the upper and lower CIs for $\text{acc}(h)$ and $\Phi_{\text{fair}}(h)$ respectively,

$$\mathbb{P}(\text{acc}(h) \leq U_{\text{acc}}^h) \geq 1 - \alpha/2 \quad \text{and} \quad \mathbb{P}(\Phi_{\text{fair}}(h) \geq L_{\text{fair}}^h) \geq 1 - \alpha/2.$$

Then, using an application of union bounds, we get that

$$\mathbb{P}(\text{acc}(h) \leq U_{\text{acc}}^h, \Phi_{\text{fair}}(h) \geq L_{\text{fair}}^h) \geq 1 - \mathbb{P}(\text{acc}(h) > U_{\text{acc}}^h) - \mathbb{P}(\Phi_{\text{fair}}(h) < L_{\text{fair}}^h)$$
$$\geq 1 - \alpha/2 - \alpha/2 = 1 - \alpha.$$

Then, using the fact that $\Delta(h) = \Phi_{\text{fair}}(h) - \tau_{\text{fair}}^*(\text{acc}(h))$, we get that

$$1 - \alpha \leq \mathbb{P}(\text{acc}(h) \leq U_{\text{acc}}^h, \Phi_{\text{fair}}(h) \geq L_{\text{fair}}^h) = \mathbb{P}(\text{acc}(h) \leq U_{\text{acc}}^h, \tau_{\text{fair}}^*(\text{acc}(h)) + \Delta(h) \geq L_{\text{fair}}^h)$$
$$\leq \mathbb{P}(\text{acc}(h) \leq U_{\text{acc}}^h, \tau_{\text{fair}}^*(U_{\text{acc}}^h) + \Delta(h) \geq L_{\text{fair}}^h)$$
$$\leq \mathbb{P}(\tau_{\text{fair}}^*(U_{\text{acc}}^h) \geq L_{\text{fair}}^h - \Delta(h)),$$

where in the second last inequality above, we use the fact that $\tau_{\text{fair}}^* : [0,1] \to [0,1]$ is a monotonically increasing function. □

## A.2 Asymptotic convergence of $\Delta(h)$

In this Section, we provide the formal statement for Theorem 3.4 along with the assumptions required for this result.

**Assumption A.1.** $\tau_{\text{fair}}^*$ is $L$-Lipschitz.

**Assumption A.2.** Let $\mathcal{R}_{|\mathcal{D}_{\text{tr}}|}(\mathcal{H})$ denote the Rademacher complexity of the classifier family $\mathcal{H}$, where $|\mathcal{D}_{\text{tr}}|$ is the number of training examples. We assume that there exists $C \geq 0$ and $\gamma \leq 1/2$ such that $\mathcal{R}_{|\mathcal{D}_{\text{tr}}|}(\mathcal{H}) \leq C |\mathcal{D}_{\text{tr}}|^{-\gamma}$.

It is worth noting that Assumption A.2, which was also used in [1, Theorem 2], holds for many classifier families with $\gamma = 1/2$, including norm-bounded linear functions, neural networks and classifier families with bounded VC dimension [21, 6].

**Theorem A.3.** *Let $\widehat{\Phi_{\text{fair}}}(h'), \widehat{\text{acc}}(h')$ denote the fairness violation and accuracy metrics for model $h'$ evaluated on training data $\mathcal{D}_{\text{tr}}$ and define*

$$h := \arg\min_{h' \in \mathcal{H}} \widehat{\Phi_{\text{fair}}}(h') \text{ subject to } \widehat{\text{acc}}(h') \geq \delta.$$

*Then, under Assumptions A.1 and A.2, we have that with probability at least $1 - \eta$, $\Delta(h) \leq \widetilde{O}(|\mathcal{D}_{\text{tr}}|^{-\gamma})$, where $\widetilde{O}(\cdot)$ suppresses polynomial dependence on $\log(1/\eta)$.*

*Proof of Theorem A.3.* Let

$$\epsilon := 4\,\mathcal{R}_{|\mathcal{D}_{\text{tr}}|}(\mathcal{H}) + \frac{4}{\sqrt{|\mathcal{D}_{\text{tr}}|}} + 2\sqrt{\frac{\log(8/\eta)}{2\,|\mathcal{D}_{\text{tr}}|}}.$$

Next, we define

$$h^* := \arg\min_{h \in \mathcal{H}} \Phi_{\text{fair}}(h) \quad \text{subject to} \quad \text{acc}(h) \geq \text{acc}(h) + \epsilon.$$

Then, we have that

$$\begin{aligned}
\Delta(h) =& \Phi_{\text{fair}}(h) - \tau^*_{\text{fair}}(\text{acc}(h)) \\
=& \Phi_{\text{fair}}(h) - \tau^*_{\text{fair}}(\text{acc}(h) + \epsilon) + \tau^*_{\text{fair}}(\text{acc}(h) + \epsilon) - \tau^*_{\text{fair}}(\text{acc}(h)) \\
=& \Phi_{\text{fair}}(h) - \Phi_{\text{fair}}(h^*) + \tau^*_{\text{fair}}(\text{acc}(h) + \epsilon) - \tau^*_{\text{fair}}(\text{acc}(h)).
\end{aligned}$$

We know using Assumption A.1 that

$$\tau^*_{\text{fair}}(\text{acc}(h) + \epsilon) - \tau^*_{\text{fair}}(\text{acc}(h)) \leq L\,\epsilon$$

Moreover,

$$\begin{aligned}
\Phi_{\text{fair}}(h) - \Phi_{\text{fair}}(h^*) =& \widehat{\Phi_{\text{fair}}}(h) - \widehat{\Phi_{\text{fair}}}(h^*) + \Phi_{\text{fair}}(h) - \widehat{\Phi_{\text{fair}}}(h) + \widehat{\Phi_{\text{fair}}}(h^*) - \Phi_{\text{fair}}(h^*) \\
\leq& \widehat{\Phi_{\text{fair}}}(h) - \widehat{\Phi_{\text{fair}}}(h^*) + 2\max_{h' \in \mathcal{H}} |\Phi_{\text{fair}}(h') - \widehat{\Phi_{\text{fair}}}(h')|.
\end{aligned}$$

Putting the two together, we get that

$$\Delta(h) \leq \widehat{\Phi_{\text{fair}}}(h) - \widehat{\Phi_{\text{fair}}}(h^*) + 2\max_{h' \in \mathcal{H}} |\Phi_{\text{fair}}(h') - \widehat{\Phi_{\text{fair}}}(h')| + L\epsilon. \tag{5}$$

Next, we consider the term $\widehat{\Phi_{\text{fair}}}(h) - \widehat{\Phi_{\text{fair}}}(h^*)$. First, we observe that

$$\text{acc}(h^*) \geq \text{acc}(h) + \epsilon \geq \widehat{\text{acc}}(h) - |\widehat{\text{acc}}(h) - \text{acc}(h)| + \epsilon \geq \delta + \epsilon - |\widehat{\text{acc}}(h) - \text{acc}(h)|.$$

Next, using [1, Lemma 4] with $g(X, A, Y) = \mathbb{1}(h(X) = Y)$, we get that with probability at least $1 - \eta/4$, we have that

$$|\text{acc}(h) - \widehat{\text{acc}}(h)| \leq 2\,\mathcal{R}_{|\mathcal{D}_{\text{tr}}|}(\mathcal{H}) + \frac{2}{\sqrt{|\mathcal{D}_{\text{tr}}|}} + \sqrt{\frac{\log(8/\eta)}{2\,|\mathcal{D}_{\text{tr}}|}} = \epsilon/2.$$

This implies that with probability at least $1 - \eta/4$,

$$\text{acc}(h^*) \geq \delta + \epsilon/2,$$

and hence,

$$\delta + \epsilon/2 \leq \text{acc}(h^*) \leq \widehat{\text{acc}}(h^*) + |\text{acc}(h^*) - \widehat{\text{acc}}(h^*)|.$$

Again, using [1, Lemma 4] with $g(X, A, Y) = \mathbb{1}(h^*(X) = Y)$, we get that with probability at least $1 - \eta/4$, we have that

$$|\text{acc}(h^*) - \widehat{\text{acc}}(h^*)| \leq 2\,\mathcal{R}_{|\mathcal{D}_{\text{tr}}|}(\mathcal{H}) + \frac{2}{\sqrt{|\mathcal{D}_{\text{tr}}|}} + \sqrt{\frac{\log(8/\eta)}{2\,|\mathcal{D}_{\text{tr}}|}} = \epsilon/2.$$

Finally, putting this together using union bounds, we get that with probability at least $1 - \eta/2$,

$$\delta + \epsilon/2 \le \mathrm{acc}(h^*) \le \widehat{\mathrm{acc}(h^*)} + |\mathrm{acc}(h^*) - \widehat{\mathrm{acc}(h^*)}| \le \widehat{\mathrm{acc}(h^*)} + \epsilon/2,$$

and hence,

$$\widehat{\mathrm{acc}(h^*)} \ge \delta.$$

Using the definition of $h$, we have that $\widehat{\mathrm{acc}(h^*)} \ge \delta \implies \widehat{\Phi_{\mathrm{fair}}(h)} \le \widehat{\Phi_{\mathrm{fair}}(h^*)}$. Therefore, with probability at least $1 - \eta/2$,

$$\widehat{\Phi_{\mathrm{fair}}(h)} - \widehat{\Phi_{\mathrm{fair}}(h^*)} \le 0. \tag{6}$$

Next, to bound the term $|\Phi_{\mathrm{fair}}(h) - \widehat{\Phi_{\mathrm{fair}}(h)}|$ above, we consider a general formulation for the fairness violation $\Phi_{\mathrm{fair}}$, also presented in [1],

$$\Phi_{\mathrm{fair}}(h) = |\Phi_{\mathrm{fair}}^{\pm}(h)| \quad \text{where,} \quad \Phi_{\mathrm{fair}}^{\pm}(h) := \sum_{j=1}^{m} \underbrace{\mathbb{E}[g_j(X, A, Y, h(X)) \mid \mathcal{E}_j]}_{=: \Phi_j}$$

where $m \ge 1$, $g_j$ are some known functions and $\mathcal{E}_j$ are events with positive probability defined with respect to $(X, A, Y)$.

Using this, we note that for any $h' \in \mathcal{H}$

$$
\begin{aligned}
|\Phi_{\mathrm{fair}}(h') - \widehat{\Phi_{\mathrm{fair}}(h')}| &= \left| |\Phi_{\mathrm{fair}}^{\pm}(h')| - |\widehat{\Phi_{\mathrm{fair}}^{\pm}(h')}| \right| \\
&\le |\Phi_{\mathrm{fair}}^{\pm}(h') - \widehat{\Phi_{\mathrm{fair}}^{\pm}(h')}| \\
&= \left| \sum_{j=1}^{m} \mathbb{E}[g_j(X, A, Y, h'(X)) \mid \mathcal{E}_j] - \widehat{\mathbb{E}}[g_j(X, A, Y, h'(X)) \mid \mathcal{E}_j] \right| \\
&\le \sum_{j=1}^{m} |\mathbb{E}[g_j(X, A, Y, h'(X)) \mid \mathcal{E}_j] - \widehat{\mathbb{E}}[g_j(X, A, Y, h'(X)) \mid \mathcal{E}_j]|.
\end{aligned}
$$

Let $p_j^* := \mathbb{P}(\mathcal{E}_j)$. Then, from [1, Lemma 6], we have that if $|\mathcal{D}_{\mathrm{tr}}| p_j^* \ge 8 \log(4\, m/\eta)$, then with probability at least $1 - \eta/2m$, we have that

$$|\mathbb{E}[g_j(X, A, Y, h'(X)) \mid \mathcal{E}_j] - \widehat{\mathbb{E}}[g_j(X, A, Y, h'(X)) \mid \mathcal{E}_j]| \le 2\, R_{|\mathcal{D}_{\mathrm{tr}}| p_j^*/2}(\mathcal{H}) + 2\sqrt{\frac{2}{|\mathcal{D}_{\mathrm{tr}}| p_j^*}} + \sqrt{\frac{\log(8\, m/\eta)}{|\mathcal{D}_{\mathrm{tr}}| p_j^*}}$$

$$= \widetilde{O}(|\mathcal{D}_{\mathrm{tr}}|^{-\gamma}).$$

A straightforward application of the union bounds yields that if $|\mathcal{D}_{\mathrm{tr}}| p_j^* \ge 8 \log(4\, m/\eta)$ for all $j \in \{1, \ldots, m\}$, then with probability at least $1 - \eta/2$, we have that

$$\sum_{j=1}^{m} |\mathbb{E}[g_j(X, A, Y, h'(X)) \mid \mathcal{E}_j] - \widehat{\mathbb{E}}[g_j(X, A, Y, h'(X)) \mid \mathcal{E}_j]| \le \widetilde{O}(|\mathcal{D}_{\mathrm{tr}}|^{-\gamma}).$$

Therefore, in this case for any $h' \in \mathcal{H}$, we have that with probability at least $1 - \eta/2$,

$$|\Phi_{\mathrm{fair}}(h') - \widehat{\Phi_{\mathrm{fair}}(h')}| \le \widetilde{O}(|\mathcal{D}_{\mathrm{tr}}|^{-\gamma}),$$

and therefore,

$$2 \max_{h' \in \mathcal{H}} |\Phi_{\mathrm{fair}}(h') - \widehat{\Phi_{\mathrm{fair}}(h')}| \le \widetilde{O}(|\mathcal{D}_{\mathrm{tr}}|^{-\gamma}). \tag{7}$$

Finally, putting Eq. (5), Eq. (6) and Eq. (7) together using union bounds, we get that with probability at least $1 - \eta$, we have that

$$\Delta(h) \le \widehat{\Phi_{\mathrm{fair}}(h)} - \widehat{\Phi_{\mathrm{fair}}(h^*)} + 2 \max_{h' \in \mathcal{H}} |\Phi_{\mathrm{fair}}(h') - \widehat{\Phi_{\mathrm{fair}}(h')}| + L\epsilon \le \widetilde{O}(|\mathcal{D}_{\mathrm{tr}}|^{-\gamma}).$$

$\square$

# B    Constructing the confidence intervals on $\Phi_{\textbf{fair}}(h)$

In this section, we outline methodologies of obtaining confidence intervals for a fairness violation $\Phi_{\text{fair}}$. Specifically, given a model $h \in \mathcal{H}$, with $h : \mathcal{X} \to \mathcal{Y}$ and $\alpha \in (0,1)$, we outline how to find $C_{\text{fair}}^{\alpha}$ which satisfies,

$$\mathbb{P}(\Phi_{\text{fair}}(h) \in C_{\text{fair}}^{\alpha}) \geq 1 - \alpha. \tag{8}$$

Similar to [1] we express the fairness violation $\Phi_{\text{fair}}$ as:

$$\Phi_{\text{fair}}(h) = |\Phi_{\text{fair}}^{\pm}(h)| \quad \text{where,} \quad \Phi_{\text{fair}}^{\pm}(h) := \sum_{j=1}^{m} \underbrace{\mathbb{E}[g_j(X, A, Y, h(X)) \mid \mathcal{E}_j]}_{=:\Phi_j}$$

where $m \geq 1$, $g_j$ are some known functions and $\mathcal{E}_j$ are events with positive probability defined with respect to $(X, A, Y)$. For example, when considering the demographic parity (DP), i.e. $\Phi_{\text{fair}} = \Phi_{\text{DP}}$, we have $m = 2$, with $g_1(X, A, Y, h(X)) = h(X)$, $\mathcal{E}_1 = \{A = 1\}$, $g_2(X, A, Y, h(X)) = -h(X)$ and $\mathcal{E}_2 = \{A = 0\}$. Moreover, as shown in [1], the commonly used fairness metrics like Equalized Odds (EO) and Equalized Opportunity (EOP) can also be expressed in similar forms.

Our methodology of constructing CIs on $\Phi_{\text{fair}}(h)$ involves first constructing intervals $C_{\text{fair}}^{\alpha,\pm}$ on $\Phi_{\text{fair}}^{\pm}(h)$ satisfying:

$$\mathbb{P}(\Phi_{\text{fair}}^{\pm}(h) \in C_{\text{fair}}^{\alpha,\pm}) \geq 1 - \alpha. \tag{9}$$

Once we have a $C_{\text{fair}}^{\alpha,\pm}$, the confidence interval $C_{\text{fair}}^{\alpha}$ satisfying Eq. (8) can simply be constructed as:

$$C_{\text{fair}}^{\alpha} = \{|x| \, : \, x \in C_{\text{fair}}^{\alpha,\pm}\}.$$

In what follows, we outline two different ways of constructing the confidence intervals $C_{\text{fair}}^{\alpha,\pm}$ on $\Phi_{\text{fair}}^{\pm}(h)$ satisfying Eq. (9).

## B.1    Separately constructing CIs on $\Phi_j$

One way to obtain intervals on $\Phi_{\text{fair}}^{\pm}(h)$ would be to separately construct confidence intervals on $\Phi_j$, denoted by $C_j^{\alpha}$, which satisfies the joint guarantee

$$\mathbb{P}\left(\cap_{j=1}^{m}\{\Phi_j \in C_j^{\alpha}\}\right) \geq 1 - \alpha. \tag{10}$$

Given such set of confidence intervals $\{C_j^{\alpha}\}_{j=1}^{m}$ which satisfy Eqn. Eq. (10), we can obtain the confidence intervals on $\Phi_{\text{fair}}^{\pm}(h)$ by using the fact that

$$\mathbb{P}\left(\Phi_{\text{fair}}^{\pm}(h) \in \sum_{i=1}^{m} C_j^{\alpha}\right) \geq 1 - \alpha.$$

Where, the notation $\sum_{i=1}^{m} C_j^{\alpha}$ denotes the set $\{\sum_{i=1}^{m} x_i \, : \, x_i \in C_i^{\alpha}\}$. One naïve way to obtain such $\{C_j^{\alpha}\}_{j=1}^{m}$ which satisfy Eq. (10) is to use the union bounds, i.e., if $C_j^{\alpha}$ are chosen such that

$$\mathbb{P}(\Phi_j \in C_j^{\alpha}) \geq 1 - \alpha/m,$$

then, we have that

$$\mathbb{P}\left(\cap_{j=1}^{m}\{\Phi_j \in C_j^{\alpha}\}\right) = 1 - \mathbb{P}(\cup_{j=1}^{m}\{\Phi_j \in C_j^{\alpha}\}^c)$$

$$\geq 1 - \sum_{i=1}^{m} \mathbb{P}(\{\Phi_j \in C_j^{\alpha}\}^c)$$

$$\geq 1 - \sum_{i=1}^{m}(1 - (1 - \alpha/m)) = 1 - \alpha.$$

Here, for an event $\mathcal{E}$, we use $\mathcal{E}^c$ to denote the complement of the event. This methodology therefore reduces the problem of finding confidence intervals on $\Phi_{\text{fair}}^{\pm}(h)$ to finding confidence intervals on $\Phi_j$ for $j \in \{1, \ldots, m\}$. Now note that $\Phi_j$ are all expectations and we can use standard methodologies to construct confidence intervals on an expectation. We explicitly outline how to do this in Section B.3.

**Remark** The methodology outlined above provides confidence intervals with valid finite sample coverage guarantees. However, this may come at the cost of more conservative confidence intervals. One way to obtain less conservative confidence intervals while retaining the coverage guarantees would be to consider alternative ways of obtaining confidence intervals which do not require constructing the CIs separately on $\Phi_j$. We outline one such methodology in the next section.

## B.2 Using subsampling to construct the CIs on $\Phi_{\text{fair}}^{\pm}$ directly

Here, we outline how we can avoid having to use union bounds when constructing the confidence intervals on $\Phi_{\text{fair}}^{\pm}$. Let $\mathcal{D}_j$ denote the subset of data $\mathcal{D}_{\text{cal}}$, for which the event $\mathcal{E}_j$ is true. In the case where the events $\mathcal{E}_j$ are all mutually exclusive and hence $\mathcal{D}_j$ are all disjoint subsets of data (which is true for DP, EO and EOP), we can also construct these intervals by randomly sampling without replacement datapoints $(x_i^{(j)}, a_i^{(j)}, y_i^{(j)})$ from $\mathcal{D}_j$ for $i \leq l := \min_{k \leq m} |\mathcal{D}_k|$. We use the fact that

$$\widehat{\Phi_{\text{fair}}^{\pm}}(h) = \frac{1}{l} \sum_{i=1}^{l} \sum_{j=1}^{m} g_j(x_i^{(j)}, a_i^{(j)}, y_i^{(j)}, h(x_i^{(j)}))$$

is an unbiased estimator of $\Phi_{\text{fair}}^{\pm}(h)$. Moreover, since $\mathcal{D}_j$ are all disjoint datasets, the datapoints $(x_i^{(j)}, a_i^{(j)}, y_i^{(j)})$ are all independent across different values of $j$, and therefore, $\sum_{j=1}^{m} g_j(x_i^{(j)}, a_i^{(j)}, y_i^{(j)}, h(x_i^{(j)}))$ are i.i.d.. In other words,

$$\widehat{\Phi_{\text{fair}}^{\pm}}(h) = \frac{1}{l} \sum_{i=1}^{l} \phi_i \quad \text{where,} \quad \phi_i := \sum_{j=1}^{m} g_j(x_i^{(j)}, a_i^{(j)}, y_i^{(j)}, h(x_i^{(j)}))$$

and $\phi_i$ are all i.i.d. samples and unbiased estimators of $\Phi_{\text{fair}}^{\pm}(h)$. Therefore, like in the previous section, our problem reduces to constructing CIs on an expectation term (i.e. $\Phi_{\text{fair}}^{\pm}(h)$), using i.i.d. unbiased samples (i.e. $\phi_i$) and we can use standard methodologies to construct these intervals.

**Benefit of this methodology** This methodology no longer requires us to separately construct confidence intervals over $\Phi_j$ and combine them using union bounds (for example). Therefore, intervals obtained using this methodology may be less conservative than those obtained by separately constructing confidence intervals over $\Phi_j$.

**Limitation of this methodology** For each subset of data $\mathcal{D}_j$, we can use at most $l := \min_{k \leq m} |\mathcal{D}_k|$ data points to construct the confidence intervals. Therefore, in cases where $l$ is very small, we may end up discarding a big proportion of the calibration data which could in turn lead to loose intervals.

## B.3 Constructing CIs on expectations

Here, we outline some standard techniques used to construct CIs on the expectation of a random variable. These techniques can then be used to construct CIs on $\Phi_{\text{fair}}(h)$ (using either of the two methodologies outlined above) as well as on $\text{acc}(h)$. In this section, we restrict ourselves to constructing upper CIs. Lower CIs can be constructed analogously.

Given dataset $\{Z_i : 1 \leq i \leq n\}$, our goal in this section is to construct upper CIs on $\mathbb{E}[Z]$ which satisfies

$$\mathbb{P}(\mathbb{E}[Z] \leq U^{\alpha}) \geq 1 - \alpha.$$

**Hoeffding's inequality** We can use Hoeffding's inequality to construct these intervals $U^{\alpha}$ as formalised in the following result:

**Lemma B.1** (Hoeffding's inequality). *Let $Z_i \in [0, 1]$, $1 \leq i \leq n$ be i.i.d. samples with mean $\mathbb{E}[Z]$. Then,*

$$\mathbb{P}\left(\mathbb{E}[Z] \leq \frac{1}{n} \sum_{i=1}^{n} Z_i + \sqrt{\frac{1}{2n} \log \frac{1}{\alpha}}\right) \geq 1 - \alpha.$$

**Bernstein's inequality** Bernstein's inequality provides a powerful tool for bounding the tail probabilities of the sum of independent, bounded random variables. Specifically, for a sum $\sum_{i=1}^{n} Z_i$

comprised of $n$ independent random variables $Z_i$ with $Z_i \in [0, B]$, each with a maximum variance of $\sigma^2$, and for any $t > 0$, the inequality states that

$$\mathbb{P}\left(\mathbb{E}\left[\sum_{i=1}^{n} Z_i\right] - \sum_{i=1}^{n} Z_i > t\right) \leq \exp\left(-\frac{t^2}{2\sigma^2 + \frac{2}{3}tB}\right),$$

where $B$ denotes an upper bound on the absolute value of each random variable. Re-arranging the above, we get that

$$\mathbb{P}\left(\mathbb{E}[Z] < \frac{1}{n}\left(\sum_{i=1}^{n} Z_i + t\right)\right) \geq 1 - \exp\left(-\frac{t^2}{2\sigma^2 + \frac{2}{3}tB}\right).$$

This allows us to construct upper CIs on $\mathbb{E}[Z]$.

**Central Limit Theorem** The Central Limit Theorem (CLT) [26] serves as a cornerstone in statistics for constructing confidence intervals around sample means, particularly when the sample size is substantial. The theorem posits that, for a sufficiently large sample size, the distribution of the sample mean will closely resemble a normal (Gaussian) distribution, irrespective of the original population's distribution. This Gaussian nature of the sample mean empowers us to form confidence intervals for the population mean using the normal distribution's characteristics.

Given $Z_1, Z_2, \ldots, Z_n$ as $n$ independent and identically distributed (i.i.d.) random variables with mean $\mu$ and variance $\sigma^2$, the sample mean $\bar{Z}$ approximates a normal distribution with mean $\mu$ and variance $\sigma^2/n$ for large $n$. An upper $(1 - \alpha)$ confidence interval for $\mu$ is thus:

$$U^\alpha = \bar{Z} + z_{\alpha/2}\frac{\sigma}{\sqrt{n}}$$

where $z_{\alpha/2}$ represents the critical value from the standard normal distribution corresponding to a cumulative probability of $1 - \alpha$.

**Bootstrap Confidence Intervals** Bootstrapping, introduced by [17], offers a non-parametric approach to estimate the sampling distribution of a statistic. The method involves repeatedly drawing samples (with replacement) from the observed data and recalculating the statistic for each resample. The resulting empirical distribution of the statistic across bootstrap samples forms the basis for confidence interval construction.

Given a dataset $Z_1, Z_2, \ldots, Z_n$, one can produce $B$ bootstrap samples by selecting $n$ observations with replacement from the original data. For each of these samples, the statistic of interest (for instance, the mean) is determined, yielding $B$ bootstrap estimates. An upper $(1 - \alpha)$ bootstrap confidence interval for $\mathbb{E}[Z]$ is given by:

$$U^\alpha = \bar{Z} + (z_\alpha^* - \bar{Z})$$

with $z_\alpha^*$ denoting the $\alpha$-quantile of the bootstrap estimates. It's worth noting that there exist multiple methods to compute bootstrap confidence intervals, including the basic, percentile, and bias-corrected approaches, and the method described above serves as a general illustration.

## C Sensitivity analysis for $\Delta(h)$

Recall from Proposition 3.3 that the lower confidence intervals for $\tau_{\text{fair}}^*$ include a $\Delta(h)$ term which is defined as

$$\Delta(h) := \Phi_{\text{fair}}(h) - \tau_{\text{fair}}^*(\text{acc}(h)) \geq 0.$$

In other words, $\Delta(h)$ quantifies how 'far' the fairness loss of classifier $h$ (i.e. $\Phi_{\text{fair}}(h)$) is from the minimum attainable fairness loss for classifiers with accuracy $\text{acc}(h)$, (i.e. $\tau_{\text{fair}}^*(\text{acc}(h))$). This quantity is unknown in general and therefore, a practical strategy of obtaining lower confidence intervals on $\tau_{\text{fair}}^*(\psi)$ may involve positing values for $\Delta(h)$ which encode our belief on how close the fairness loss $\Phi_{\text{fair}}(h)$ is to $\tau_{\text{fair}}^*(\text{acc}(h))$. For example, when we assume that the classifier $h$ achieves the optimal accuracy-fairness tradeoff, i.e. $\Phi_{\text{fair}}(h) = \tau_{\text{fair}}^*(\text{acc}(h))$ then $\Delta(h) = 0$.

However, the assumption $\Phi_{\text{fair}}(h) = \tau_{\text{fair}}^*(\text{acc}(h))$ may not hold in general because we only have a finite training dataset and consequently the empirical loss minimisation may not yield the optima to the true expected loss. Moreover, the regularised loss used in training $h$ is a surrogate loss which

approximates the solution to the constrained minimisation problem in Eq. (1). This means that optimising this regularised loss is not guaranteed to yield the optimal classifier which achieves the optimal fairness $\tau^*_{\text{fair}}(\text{acc}(h))$. Therefore, to incorporate any belief on the sub-optimality of the classifier $h$, we may consider conducting sensitivity analyses to plausibly quantify $\Delta(h)$.

Let $h_\theta : \mathcal{X} \times \Lambda \to \mathbb{R}$ be the YOTO model. Our strategy for sensitivity analysis involves training multiple standard models $\mathcal{M} := \{h^{(1)}, h^{(2)}, \ldots, h^{(k)}\} \subseteq \mathcal{H}$ by optimising the regularised losses for few different choices of $\lambda$.

$$\mathcal{L}_\lambda(\theta) = \mathbb{E}[l_{\text{CE}}(h_\theta(X), Y)] + \lambda \, \mathcal{L}_{\text{fair}}(h_\theta).$$

Importantly, we do not require covering the full range of $\lambda$ values when training separate models $\mathcal{M}$, and our methodology remains valid even when $\mathcal{M}$ is a single model. Next, let $h^*_\lambda \in \mathcal{M} \cup \{h_\theta(\cdot, \lambda)\}$ be such that

$$h^*_\lambda = \underset{h' \in \mathcal{M} \cup \{h_\theta(\cdot, \lambda)\}}{\arg\min} \widehat{\Phi_{\text{fair}}}(h') \quad \text{subject to} \quad \widehat{\text{acc}}(h') \geq \widehat{\text{acc}}(h_\theta(\cdot, \lambda)). \tag{11}$$

Here, $\widehat{\Phi_{\text{fair}}}$ and $\widehat{\text{acc}}$ denote the finite sample estimates of the fairness loss and model accuracy respectively. We treat the model $h^*_\lambda$ as a model which attains the optimum trade-off when estimating subject to the constraint $\text{acc}(h) \geq \text{acc}(h_\theta(\cdot, \lambda))$. Specifically, we use the maximum empirical error $\max_\lambda \widehat{\Delta}(h_\theta(\cdot, \lambda))$ as a plausible surrogate value for $\Delta(h_\theta(\cdot, \lambda'))$, where $\widehat{\Delta}(h_\theta(\cdot, \lambda)) := \widehat{\Phi_{\text{fair}}}(h_\theta(\cdot, \lambda)) - \widehat{\Phi_{\text{fair}}}(h^*_\lambda) \geq 0$, i.e., we posit for any $\lambda' \in \Lambda$

$$\Delta(h_\theta(\cdot, \lambda')) \leftarrow \max_{\lambda \in \Lambda} \widehat{\Delta}(h_\theta(\cdot, \lambda)) \qquad \text{where,} \qquad \widehat{\Delta}(h_\theta(\cdot, \lambda)) := \widehat{\Phi_{\text{fair}}}(h_\theta(\cdot, \lambda)) - \widehat{\Phi_{\text{fair}}}(h^*_\lambda).$$

Next, we can use this posited value of $\Delta(h_\theta(\cdot, \lambda'))$ to construct the lower confidence interval using the following corollary of Proposition 3.3:

**Corollary C.1.** *Consider the YOTO model $h_\theta : \mathcal{X} \times \Lambda \to \mathbb{R}$. Given $\lambda_0 \in \Lambda$, let $U^h_{\text{acc}}, L^h_{\text{fair}} \in [0, 1]$ be such that*

$$\mathbb{P}(\text{acc}(h_\theta(\cdot, \lambda_0)) \leq U^h_{\text{acc}}) \geq 1 - \alpha/2 \quad \text{and} \quad \mathbb{P}(\Phi_{\text{fair}}(h_\theta(\cdot, \lambda_0)) \geq L^h_{\text{fair}}) \geq 1 - \alpha/2.$$

*Then, we have that $\mathbb{P}(\tau^*_{\text{fair}}(U^h_{\text{acc}}) \geq L^h_{\text{fair}} - \Delta(h_\theta(\cdot, \lambda_0))) \geq 1 - \alpha$.*

This result shows that if the goal is to construct lower confidence intervals on $\tau^*_{\text{fair}}(\psi)$ and we obtain that $\psi \geq U^h_{\text{acc}}$, then using the monotonicity of $\tau^*_{\text{fair}}$ we have that $\tau^*_{\text{fair}}(\psi) \geq \tau^*_{\text{fair}}(U^h_{\text{acc}})$. Therefore the interval $[L^h_{\text{fair}} - \Delta(h_\theta(\cdot, \lambda_0))), 1]$ serves as a lower confidence interval for $\tau^*_{\text{fair}}(\psi)$.

**When YOTO satisfies Pareto optimality,** $\Delta(h_\theta(\cdot, \lambda)) \to 0$ **as** $|\mathcal{D}_{\text{cal}}| \to \infty$**:** Here, we show that in the case when YOTO achieves the optimal trade-off, then our sensitivity analysis leads to $\Delta(h_\theta(\cdot, \lambda)) = 0$ as the calibration data size increases for all $\lambda \in \Lambda$. Our arguments in this section are not formal, however, this idea can be formalised without any significant difficulty.

First, the concept of Pareto optimality (defined below) formalises the idea that YOTO achieves the optimal trade-off:

**Assumption C.2** (Pareto optimality)**.**

If for some $\lambda \in \Lambda$ and $h' \in \mathcal{H}$ we have that, $\text{acc}(h') \geq \text{acc}(h_\theta(\cdot, \lambda))$ then, $\Phi_{\text{fair}}(h') \geq \Phi_{\text{fair}}(h_\theta(\cdot, \lambda))$,

In the case when YOTO satisfies this optimality property, then it is straightforward to see that $\Delta(h_\theta(\cdot, \lambda)) = 0$ for all $\lambda \in \Lambda$. In this case, as $\mathcal{D}_{\text{cal}} \to \infty$, we get that Eq. (11) roughly becomes

$$h^*_\lambda = \underset{h' \in \mathcal{M} \cup \{h_\theta(\cdot, \lambda)\}}{\arg\min} \Phi_{\text{fair}}(h') \quad \text{subject to} \quad \text{acc}(h') \geq \text{acc}(h_\theta(\cdot, \lambda)).$$

Here, Assumption C.2 implies that $h^*_\lambda = h_\theta(\cdot, \lambda)$, and therefore

$$\widehat{\Delta}(h_\theta(\cdot, \lambda)) := \widehat{\Phi_{\text{fair}}}(h_\theta(\cdot, \lambda)) - \widehat{\Phi_{\text{fair}}}(h^*_\lambda) = 0.$$

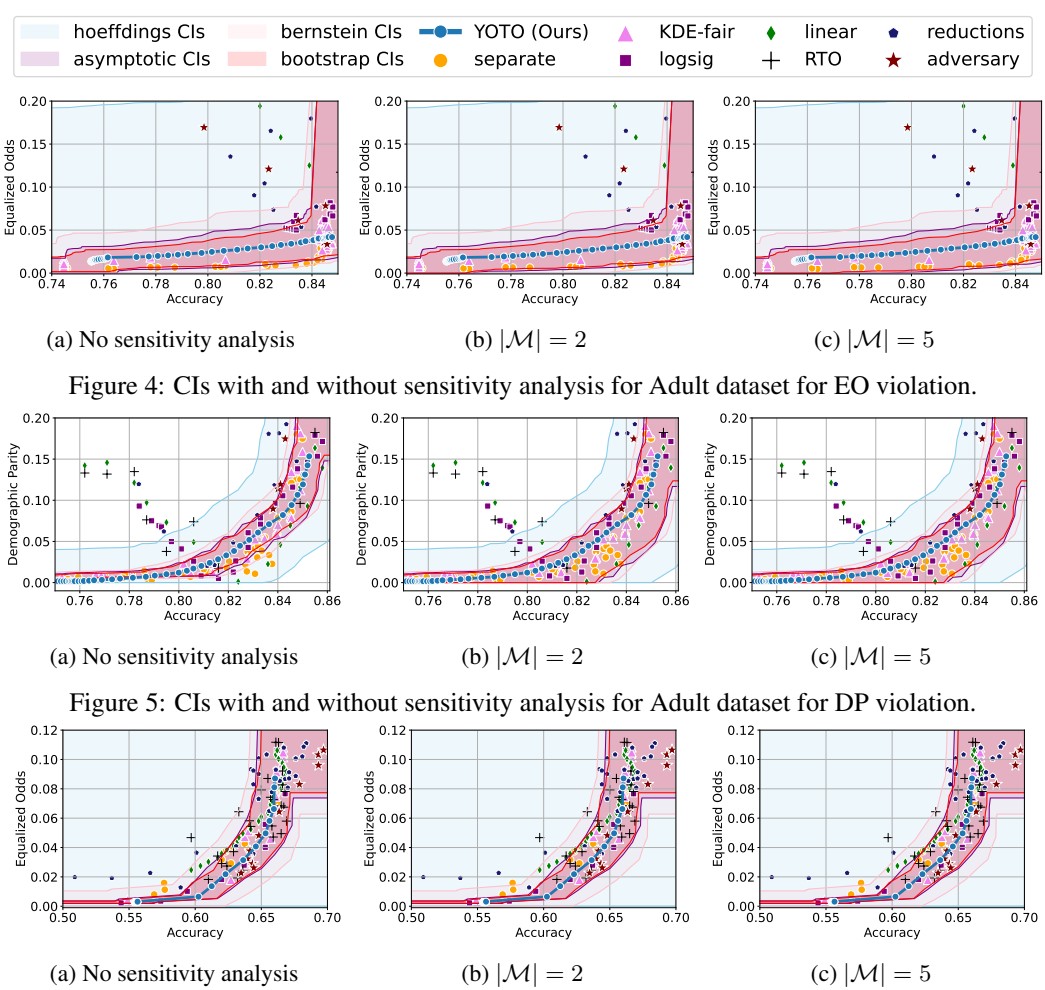

(a) No sensitivity analysis  (b) $|\mathcal{M}| = 2$  (c) $|\mathcal{M}| = 5$

Figure 4: CIs with and without sensitivity analysis for Adult dataset for EO violation.

(a) No sensitivity analysis  (b) $|\mathcal{M}| = 2$  (c) $|\mathcal{M}| = 5$

Figure 5: CIs with and without sensitivity analysis for Adult dataset for DP violation.

(a) No sensitivity analysis  (b) $|\mathcal{M}| = 2$  (c) $|\mathcal{M}| = 5$

Figure 6: CIs with and without sensitivity analysis for COMPAS dataset for EO violation. Here, sensitivity analysis has no effect on the constructed CIs as the YOTO model achieves a better empirical trade-off than the separately trained models.

**Intuition behind our sensitivity analysis procedure** Intuitively, the high-level idea behind our sensitivity analysis is that it checks if we train models separately for fixed values of $\lambda$ (i.e. models in $\mathcal{M}$), how much better do these separately trained models perform in terms of the accuracy-fairness trade-offs as compared to our YOTO model. If we find that the separately trained models achieve a better trade-off than the YOTO model for specific values of $\lambda$, then the sensitivity analysis adjusts the empirical trade-off obtained using YOTO models (using the $\widehat{\Delta}(h_\theta(\cdot, \lambda))$ term defined above). If, on the other hand, we find that the YOTO model achieves a better trade-off than the separately trained models in $\mathcal{M}$, then the sensitivity analysis has no effect on the lower confidence intervals as in this case $\widehat{\Delta}(h_\theta(\cdot, \lambda)) = 0$.

## C.1 Experimental results

Here, we include empirical results showing how the CIs constructed change as a result of our sensitivity analysis procedure. In Figures 4 and 5, we include examples of CIs where the empirical trade-off obtained using YOTO is sub-optimal. In these cases, the lower CIs obtained without sensitivity analysis (i.e. when we assume $\Delta(h_\lambda) = 0$) do not cover the empirical trade-offs for the separately trained models. However, the figures show that the sensitivity analysis procedure adjusts the lower CIs in both cases so that they encapsulate the empirical trade-offs that were not captured without sensitivity analysis.

Recall that $\mathcal{M}$ represents the set of additional separately trained models used for the sensitivity analysis. It can be seen from Figures 4 and 5 that in both cases our sensitivity analysis performs well with as little as two models (i.e. $|\mathcal{M}| = 2$), which shows that our sensitivity analysis does not come at a high computational cost.

Tables 2 and 3 contain results corresponding to these figures and show the proportion of trade-offs which lie in the three trade-off regions shown in Figure 1 with and without sensitivity analysis. It can be seen that in both tables, when $|\mathcal{M}| = 2$, the proportion of trade-offs which lie below the lower CIs (blue region in Figure 1) is negligible.

Additionally, in Figure 6 we also consider an example where YOTO achieves a better empirical trade-off than most other baselines considered, and therefore there is no need for sensitivity analysis. In this case, Figure 6 (and Table 4) show that sensitivity analysis has no effect on the CIs constructed since in this case sensitivity analysis gives us $\widehat{\Delta}(h_\theta(\cdot, \lambda)) = 0$ for $\lambda \in \Lambda$. This shows that in cases where sensitivity analysis is not needed (for example, if YOTO achieves optimal empirical trade-off), our sensitivity analysis procedure does not make the CIs more conservative.

Table 2: Results for the Adult dataset and EO fairness violation with and without sensitivity analysis: Proportion of empirical trade-offs for each baseline which lie in the three trade-off regions (using Bootstrap CIs).

| Baseline | Sub-optimal | Unlikely ($|\mathcal{M}| = 0$) | Permissible ($|\mathcal{M}| = 0$) | Unlikely ($|\mathcal{M}| = 2$) | Permissible ($|\mathcal{M}| = 2$) | Unlikely ($|\mathcal{M}| = 5$) | Permissible ($|\mathcal{M}| = 5$) |
|---|---|---|---|---|---|---|---|
| KDE-fair | 0.00 | 0.15 | 0.85 | 0.00 | 1.00 | 0.00 | 1.00 |
| RTO | 0.60 | 0.00 | 0.40 | 0.00 | 0.40 | 0.00 | 0.40 |
| adversary | 0.64 | 0.00 | 0.36 | 0.00 | 0.36 | 0.00 | 0.36 |
| linear | 0.60 | 0.00 | 0.40 | 0.00 | 0.40 | 0.00 | 0.40 |
| logsig | 0.35 | 0.00 | 0.65 | 0.00 | 0.65 | 0.00 | 0.65 |
| reductions | 0.93 | 0.00 | 0.07 | 0.00 | 0.07 | 0.00 | 0.07 |
| separate | 0.00 | 0.54 | 0.46 | 0.00 | 1.00 | 0.00 | 1.00 |

Table 3: Results for the Adult dataset and DP fairness violation with and without sensitivity analysis: Proportion of empirical trade-offs for each baseline which lie in the three trade-off regions (using Bootstrap CIs).

| Baseline | Sub-optimal | Unlikely ($|\mathcal{M}| = 0$) | Permissible ($|\mathcal{M}| = 0$) | Unlikely ($|\mathcal{M}| = 2$) | Permissible ($|\mathcal{M}| = 2$) | Unlikely ($|\mathcal{M}| = 5$) | Permissible ($|\mathcal{M}| = 5$) |
|---|---|---|---|---|---|---|---|
| KDE-fair | 0.03 | 0.10 | 0.87 | 0.00 | 0.97 | 0.00 | 0.97 |
| RTO | 0.67 | 0.00 | 0.33 | 0.00 | 0.33 | 0.00 | 0.33 |
| adversary | 0.91 | 0.00 | 0.09 | 0.00 | 0.09 | 0.00 | 0.09 |
| linear | 0.40 | 0.33 | 0.27 | 0.00 | 0.60 | 0.00 | 0.60 |
| logsig | 0.73 | 0.05 | 0.23 | 0.00 | 0.27 | 0.00 | 0.27 |
| reductions | 0.87 | 0.00 | 0.13 | 0.00 | 0.13 | 0.00 | 0.13 |
| separate | 0.03 | 0.25 | 0.71 | 0.00 | 0.97 | 0.00 | 0.97 |

Table 4: Results for the COMPAS dataset and EO fairness violation with and without sensitivity analysis: Proportion of empirical trade-offs for each baseline which lie in the three trade-off regions (using Bootstrap CIs).

| Baseline | Sub-optimal | Unlikely ($|\mathcal{M}| = 0$) | Permissible ($|\mathcal{M}| = 0$) | Unlikely ($|\mathcal{M}| = 2$) | Permissible ($|\mathcal{M}| = 2$) | Unlikely ($|\mathcal{M}| = 5$) | Permissible ($|\mathcal{M}| = 5$) |
|---|---|---|---|---|---|---|---|
| KDE-fair | 0.00 | 0.00 | 1.00 | 0.00 | 1.00 | 0.00 | 1.00 |
| RTO | 0.15 | 0.00 | 0.85 | 0.00 | 0.85 | 0.00 | 0.85 |
| adversary | 0.00 | 0.00 | 1.00 | 0.00 | 1.00 | 0.00 | 1.00 |
| linear | 0.21 | 0.00 | 0.79 | 0.00 | 0.79 | 0.00 | 0.79 |
| logsig | 0.55 | 0.00 | 0.45 | 0.00 | 0.45 | 0.00 | 0.45 |
| reductions | 0.30 | 0.00 | 0.70 | 0.00 | 0.70 | 0.00 | 0.70 |
| separate | 0.03 | 0.80 | 0.17 | 0.80 | 0.17 | 0.80 | 0.17 |

## D    Scarce sensitive attributes

Our methodology of obtaining confidence intervals on $\Phi_{\text{fair}}$ assumes access to the sensitive attributes $A$ for all data points in the held-out dataset $\mathcal{D}$. However, in practice, we may only have access to $A$ for a small proportion of the data in $\mathcal{D}$. In this case, a naïve strategy would involve constructing confidence intervals using only the data for which $A$ is available. However, since such data is scarce, the confidence intervals constructed are very loose.

Suppose that we additionally have access to a predictive model $f_{\mathcal{A}}$ which predicts the sensitive attributes $A$ using the features $X$. In this case, another simple strategy would be to simply impute the missing values of $A$, with the values $\hat{A}$ predicted using $f_{\mathcal{A}}$. However, this will usually lead to a biased estimate of the fairness violation $\Phi_{\text{fair}}(h)$, and hence is not very reliable unless the model $f_{\mathcal{A}}$ is highly accurate. In this section, we show how to get the best of both worlds, i.e. how to utilise the data with missing sensitive attributes to obtain tighter and more accurate confidence intervals on $\tau_{\text{fair}}^*(\psi)$.

Formally, we consider $\mathcal{D}_{\text{cal}} = \mathcal{D} \cup \tilde{\mathcal{D}}$ where $\mathcal{D}$ denotes a data subset of size $n$ that contains sensitive attributes (i.e. we observe $A$) and $\tilde{\mathcal{D}}$ denotes the data subset of size $N$ for which we do not observe the sensitive attributes $A$, and $N \gg n$. Additionally, for both datasets, we have predictions of the sensitive attributes made by a machine-learning algorithm $f_{\mathcal{A}} : \mathcal{X} \to \mathcal{A}$, where $f_{\mathcal{A}}(X) \approx A$. Concretely we have that $\mathcal{D} = \{(X_i, A_i, Y_i, f_{\mathcal{A}}(X_i))\}_{i=1}^n$ and $\tilde{\mathcal{D}} = \{(\tilde{X}_i, \tilde{Y}_i, f_{\mathcal{A}}(\tilde{X}_i))\}_{i=1}^N$

**High-level methodology** Our methodology is inspired by prediction-powered inference [4] which builds confidence intervals on the expected outcome $\mathbb{E}[Y]$ using data for which the true outcome $Y$ is only available for a small proportion of the dataset. In our setting, however, it is the sensitive attribute $A$ that is missing for the majority of the data (and not the outcome $Y$).

For $h \in \mathcal{H}$, let $\Phi_{\text{fair}}(h)$ be a fairness violation (such as DP or EO), and let $\widetilde{\Phi_{\text{fair}}}(h)$ be the corresponding fairness violation computed on the data distribution where $A$ is replaced by the surrogate sensitive attribute $f_{\mathcal{A}}(X)$. For example, in the case of DP violation, $\Phi_{\text{fair}}(h)$ and $\widetilde{\Phi_{\text{fair}}}(h)$ denote:

$$\Phi_{\text{fair}}(h) = |\mathbb{P}(h(X) = 1 \mid A = 1) - \mathbb{P}(h(X) = 1 \mid A = 0)|,$$
$$\widetilde{\Phi_{\text{fair}}}(h) = |\mathbb{P}(h(X) = 1 \mid f_{\mathcal{A}}(X) = 1) - \mathbb{P}(h(X) = 1 \mid f_{\mathcal{A}}(X) = 0)|.$$

We next construct the confidence intervals on $\Phi_{\text{fair}}(h)$ using the following steps:

1. Using $\mathcal{D}$, we construct intervals $C_\epsilon(\alpha; h)$ on $\epsilon(h) := \Phi_{\text{fair}}(h) - \widetilde{\Phi_{\text{fair}}}(h)$ satisfying

$$\mathbb{P}(\epsilon(h) \in C_\epsilon(\alpha; h)) \geq 1 - \alpha. \tag{12}$$

   Even though the size of $\mathcal{D}$ is small, we choose a methodology which yields tight intervals for $\epsilon(h)$ when $f_{\mathcal{A}}(X_i) = A_i$ with a high probability.

2. Next, using the dataset $\tilde{\mathcal{D}}$, we construct intervals $\tilde{C}_{\text{f}}(\alpha; h)$ on $\widetilde{\Phi_{\text{fair}}}(h)$ satisfying

$$\mathbb{P}(\widetilde{\Phi_{\text{fair}}}(h) \in \tilde{C}_{\text{f}}(\alpha; h)) \geq 1 - \alpha. \tag{13}$$

   This interval will also be tight as the size of $\tilde{\mathcal{D}}$, $N \gg n$.

Finally, using the union bound idea we combine the two confidence intervals to obtain the confidence interval for $\Phi_{\text{fair}}(h) - \widetilde{\Phi_{\text{fair}}}(h) + \widetilde{\Phi_{\text{fair}}}(h) = \Phi_{\text{fair}}(h)$. We make this precise in the following result:

**Lemma D.1.** *Let* $C_\epsilon(\alpha; h), \tilde{C}_{\text{f}}(\alpha; h)$ *be as defined in equations 12 and 13. Then, if we define* $C_{\text{fair}}^\alpha(h) = \{x + y \mid x \in C_\epsilon(\alpha; h), y \in \tilde{C}_{\text{f}}(\alpha; h)\}$, *we have that*

$$\mathbb{P}(\Phi_{\text{fair}}(h) \in C_{\text{fair}}^\alpha(h)) \geq 1 - 2\alpha.$$

When constructing the CIs over $\widetilde{\Phi_{\text{fair}}}(h)$ using imputed sensitive attributes $f_{\mathcal{A}}(X)$ in step 2 above, the prediction error of $f_{\mathcal{A}}$ introduces an error in the obtained CIs (denoted by $\epsilon(h)$). Step 1 rectifies this by constructing a CI over the incurred error $\epsilon(h)$, and therefore combining the two allows us to obtain intervals which utilise all of the available data while ensuring that the constructed CIs are well-calibrated.

**Example: Demographic parity** Having defined our high-level methodology above, we concretely demonstrate how this can be applied to the case where the fairness loss under consideration is DP. As described above, the first step involves constructing intervals on $\epsilon(h) := \Phi_{\text{fair}}(h) - \widetilde{\Phi_{\text{fair}}}(h)$ using a methodology which yields tight intervals when $f_{\mathcal{A}}(X_i) = A_i$ with a high probability. To this end, we use bootstrapping as described in Algorithm 1.

Even though bootstrapping does not provide us with finite sample coverage guarantees, it is asymptotically exact and satisfies the property that the confidence intervals are tight when $\hat{A} = A$ with a high

probability. On the other hand, concentration inequalities (such as Hoeffding's inequality) seek to construct confidence intervals individually on $\Phi_{\text{fair}}(h)$ and $\widetilde{\Phi}_{\text{fair}}(h)$ and subsequently combine them through union bounds argument, for example. In doing so, these methods do not account for how close the values of $\Phi_{\text{fair}}(h)$ and $\widetilde{\Phi}_{\text{fair}}(h)$ might be in the data.

To make this concrete, consider the example where $f_{\mathcal{A}}(X) \stackrel{\text{a.s.}}{=} A$ and hence $\Phi_{\text{fair}}(h) = \widetilde{\Phi}_{\text{fair}}(h)$. When using concentration inequalities to construct the $1 - \alpha$ confidence intervals on $\Phi_{\text{fair}}(h)$ and $\widetilde{\Phi}_{\text{fair}}(h)$, we obtain identical intervals for the two quantities, say $[l, u]$. Then, using union bounds we obtain that $\Phi_{\text{fair}}(h) - \widetilde{\Phi}_{\text{fair}}(h) \in [l - u, u - l]$ with probability at least $1 - 2\alpha$. In this case even though $\Phi_{\text{fair}}(h) - \widetilde{\Phi}_{\text{fair}}(h) = 0$, the width of the interval $[l - u, u - l]$ does not depend on the closeness of $\Phi_{\text{fair}}(h)$ and $\widetilde{\Phi}_{\text{fair}}(h)$ and therefore is not tight. Bootstrapping helps us circumvent this problem, since in this case for each resample of the data $\mathcal{D}$, the finite sample estimates $\widehat{\Phi}_{\text{fair}}(h)$ and $\widehat{\widetilde{\Phi}}_{\text{fair}}(h)$ will be equal. We outline the bootstrapping algorithm below.

---

**Algorithm 1** Bootstrapping for estimating $\epsilon(h) := \Phi_{\text{fair}}(h) - \widetilde{\Phi}_{\text{fair}}(h)$

---

**Require:** Dataset $\mathcal{D}$, number of bootstrap samples $B$, significance level $\alpha$
**Ensure:** $1 - \alpha$ confidence interval for $\epsilon(h)$
  Initialize empty array $\mathbf{v}_b$
  **for** $i = 1$ to $B$ **do**
    Draw a bootstrap sample $\mathcal{D}^*$ of size $|\mathcal{D}|$ with replacement from $\mathcal{D}$
    Compute $\widehat{\Phi}_{\text{fair}}(h)$ and $\widehat{\widetilde{\Phi}}_{\text{fair}}(h)$ on $\mathcal{D}^*$
    Compute the difference $\widehat{\epsilon(h)} := \widehat{\Phi}_{\text{fair}}(h) - \widehat{\widetilde{\Phi}}_{\text{fair}}(h)$
    Append $\widehat{\epsilon(h)}$ to $\mathbf{v}_b$
  **end for**
  Compute the $\alpha/2$ and $1 - \alpha/2$ quantiles of $\mathbf{v}_b$, denoted as $l$ and $u$
  **Return:** Confidence interval $C_{\epsilon}(\alpha; h) = [l, u]$

---

Using Algorithm 1 we construct a confidence interval $C_{\epsilon}(\alpha; h)$ on $\epsilon(h)$ of size $1 - \alpha$, which approximately satisfies Eq. (12). Next, using standard techniques we can obtain an interval $\tilde{C}_{\text{f}}(\alpha; h)$ on $\widetilde{\Phi}_{\text{fair}}(h)$ using $\tilde{\mathcal{D}}$ which satisfies Eq. (13). Like before, the interval $\tilde{C}_{\text{f}}(\alpha; h)$ is likely to be tight as we use $\tilde{\mathcal{D}}$ to construct it, which is significantly larger than $\mathcal{D}$. Finally, combining the two as shown in Lemma D.1, we obtain the confidence interval on $\Phi_{\text{fair}}(h)$.

### D.1 Experimental results

Here, we present experimental results in the setting where the sensitive attributes are missing for majority of the calibration data. Figures 7-12 show the results for different datasets and predictive models $f_{\mathcal{A}}$ with varying accuracies. Here, the empirical fairness violation values for both YOTO and separately trained models are evaluated using the true sensitive attributes over the entire calibration data.

**CIs with imputed sensitive attributes are mis-calibrated** Figures 7, 9 and 11 show results for Adult, COMPAS and CelebA datasets, where the CIs are computed by imputing the missing sensitive attributes with the predicted sensitive attributes $f_{\mathcal{A}}(X) \approx A$. The figures show that when the accuracy of $f_{\mathcal{A}}$ is below 90%, the CIs are highly miscalibrated as they do not entirely contain the empirical trade-offs for both YOTO and separately trained models.

**Our methodology corrects for the mis-calibration** In contrast, Figures 8, 10 and 12 which include the corresponding results using our methodology, show that our methodology is able to correct for the mis-calibration in CIs arising from the prediction error in $f_{\mathcal{A}}$. Even though the CIs obtained using our methodology are more conservative than those obtained by imputing the missing sensitive attributes with $f_{\mathcal{A}}(X)$, they are more well-calibrated and contain the empirical trade-offs for both YOTO and separately trained model.

**Imputing missing sensitive attributes may work when $f_{\mathcal{A}}$ has high accuracy** Finally, Figures 7c, 9c and 11c show that the CIs with imputed sensitive attributes are relatively better calibrated as the

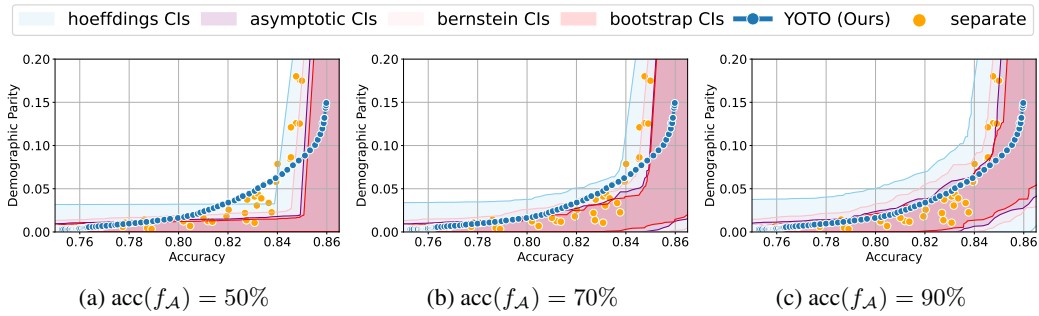

(a) $\mathrm{acc}(f_{\mathcal{A}}) = 50\%$     (b) $\mathrm{acc}(f_{\mathcal{A}}) = 70\%$     (c) $\mathrm{acc}(f_{\mathcal{A}}) = 90\%$

Figure 7: CIs obtained by imputing missing senstive attributes using $f_{\mathcal{A}}$ for Adult dataset. Here $n = 50$ and $N = 2500$.

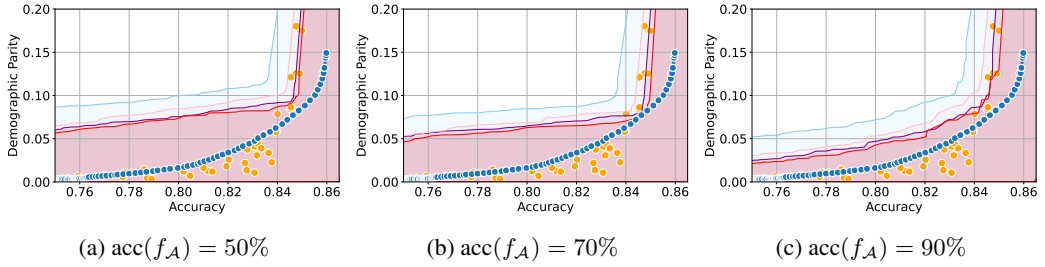

(a) $\mathrm{acc}(f_{\mathcal{A}}) = 50\%$     (b) $\mathrm{acc}(f_{\mathcal{A}}) = 70\%$     (c) $\mathrm{acc}(f_{\mathcal{A}}) = 90\%$

Figure 8: CIs were obtained using our methodology for the Adult dataset. Here $n = 50$ and $N = 2500$.

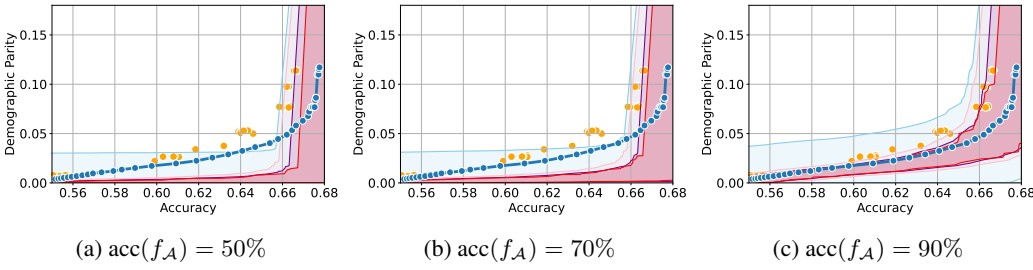

(a) $\mathrm{acc}(f_{\mathcal{A}}) = 50\%$     (b) $\mathrm{acc}(f_{\mathcal{A}}) = 70\%$     (c) $\mathrm{acc}(f_{\mathcal{A}}) = 90\%$

Figure 9: CIs obtained by imputing missing senstive attributes using $f_{\mathcal{A}}$ for COMPAS dataset. Here $n = 50$ and $N = 2000$.

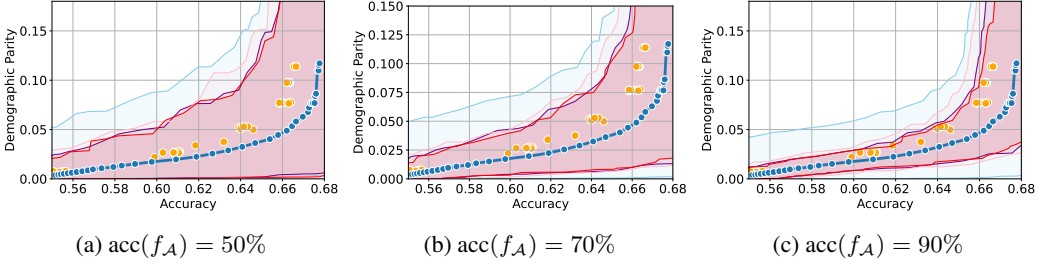

(a) $\mathrm{acc}(f_{\mathcal{A}}) = 50\%$     (b) $\mathrm{acc}(f_{\mathcal{A}}) = 70\%$     (c) $\mathrm{acc}(f_{\mathcal{A}}) = 90\%$

Figure 10: CIs obtained using our methodology for COMPAS dataset. Here $n = 50$ and $N = 2000$.

accuracy of $f_{\mathcal{A}}$ increases to 90%. In this case, the CIs with imputed sensitive attributes mostly contain empirical trade-offs. This shows that in cases where the predictive model $f_{\mathcal{A}}$ has high accuracy, it may be sufficient to impute missing sensitive attributes with $f_{\mathcal{A}}(X)$ when constructing the CIs.

## E    Accounting for the uncertainty in baseline trade-offs

In this section, we extend our methodology to also account for uncertainty in the baseline trade-offs when assessing the optimality of different baselines. Recall that our confidence intervals constructed

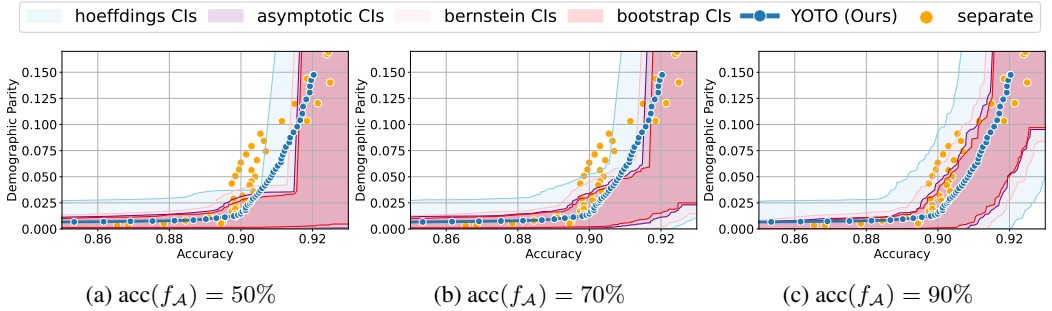

(a) $\mathrm{acc}(f_{\mathcal{A}}) = 50\%$  (b) $\mathrm{acc}(f_{\mathcal{A}}) = 70\%$  (c) $\mathrm{acc}(f_{\mathcal{A}}) = 90\%$

Figure 11: CIs obtained by imputing missing senstive attributes using $f_{\mathcal{A}}$ for CelebA dataset. Here $n = 50$ and $N = 2500$.

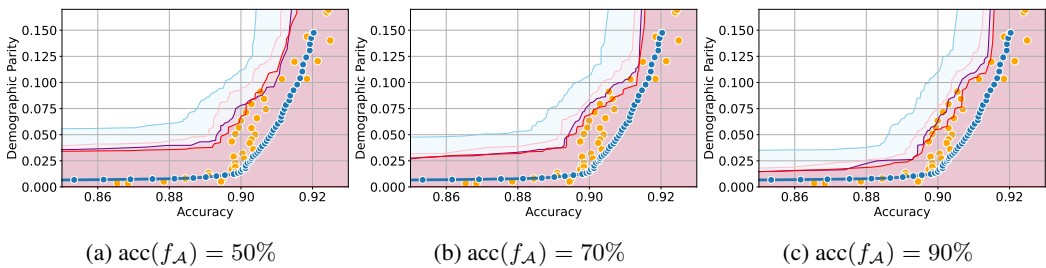

(a) $\mathrm{acc}(f_{\mathcal{A}}) = 50\%$  (b) $\mathrm{acc}(f_{\mathcal{A}}) = 70\%$  (c) $\mathrm{acc}(f_{\mathcal{A}}) = 90\%$

Figure 12: CIs obtained using our methodology for CelebA dataset. Here $n = 50$ and $N = 2500$.

on $\tau_{\mathrm{fair}}^*$ satisfy the guarantee

$$\mathbb{P}(\tau_{\mathrm{fair}}^*(\Psi) \in \Gamma_{\mathrm{fair}}^{\alpha}) \geq 1 - \alpha.$$

This means that if the accuracy-fairness tradeoff for a given model $h' \in \mathcal{H}$, $(\mathrm{acc}(h'), \Phi_{\mathrm{fair}}(h'))$, lies above the confidence intervals $\Gamma_{\mathrm{fair}}^{\alpha}$ (i.e. in the pink region in Figure 13), then we can confidently infer that the model $h'$ achieves a suboptimal trade-off. This is because we know from the probabilistic guarantee above that the optimal trade-off $(\mathrm{acc}(h'), \tau_{\mathrm{fair}}^*(\mathrm{acc}(h')))$ must lie in the intervals $\Gamma_{\mathrm{fair}}^{\alpha}$ with probability at least $1 - \alpha$.

Here, $\mathrm{acc}(h'), \Phi_{\mathrm{fair}}(h')$ denote the accuracy and fairness violations for model $h'$ on the full data distribution. However, in practice, we only have access to finite data and therefore can only compute the empirical values of accuracy and fairness violations which we denote by $\widehat{\mathrm{acc}}(h'), \widehat{\Phi_{\mathrm{fair}}}(h')$. This means that when checking if the accuracy-fairness trade-off $(\mathrm{acc}(h'), \Phi_{\mathrm{fair}}(h'))$ lies inside the confidence intervals $\Gamma_{\mathrm{fair}}^{\alpha}$, we must account for the uncertainty in the empirical estimates $\widehat{\mathrm{acc}}(h'), \widehat{\Phi_{\mathrm{fair}}}(h')$. This can be achieved by constructing confidence regions $C^{\alpha}(h')$ satisfying

$$\mathbb{P}((\mathrm{acc}(h'), \Phi_{\mathrm{fair}}(h')) \in C^{\alpha}(h')) \geq 1 - \alpha. \tag{14}$$

**Baseline's best-case accuracy-fairness trade-off** If the confidence region $C^{\alpha}(h')$ lies entirely above the confidence intervals $\Gamma_{\mathrm{fair}}^{\alpha}$ (i.e. in the pink region in Figure 13a), then using union bounds we can confidently conclude that with probability $1 - 2\alpha$ we have that the model $h'$ achieves suboptimal fairness violation, i.e.

$$\Phi_{\mathrm{fair}}(h') \geq \tau_{\mathrm{fair}}^*(\mathrm{acc}(h')).$$

This allows practitioners to confidently check if a model $h'$ is suboptimal in terms of its accuracy-fairness trade-off. This is different from simply checking if the empirical trade-off achieved by the model $(\widehat{\mathrm{acc}}(h'), \widehat{\Phi_{\mathrm{fair}}}(h'))$ lies in the permissible trade-off region (green region in Figure 1) as it also accounts for the finite-sampling uncertainty in the tradeoff achieved by model $h'$. However, this means that the criterion for flagging a baseline model $h'$ as suboptimal becomes more conservative.

Next, we show how to construct confidence regions $C^{\alpha}(h')$ satisfying Eq. (14).

**Lemma E.1.** *For a classifier $h' \in \mathcal{H}$, let $U_{\mathrm{acc}}^h, L_{\mathrm{fair}}^h$ be the upper and lower CIs on $\mathrm{acc}(h'), \Phi_{\mathrm{fair}}(h')$ respectively,*

$$\mathbb{P}(\mathrm{acc}(h') \leq U_{\mathrm{acc}}^h) \geq 1 - \alpha/2, \quad \text{and} \quad \mathbb{P}(\Phi_{\mathrm{fair}}(h') \geq L_{\mathrm{fair}}^h) \geq 1 - \alpha/2.$$

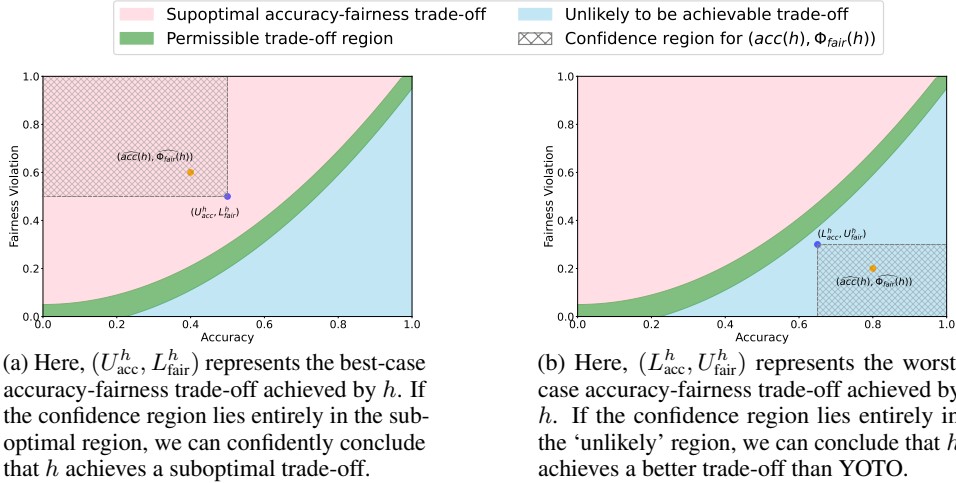

(a) Here, $(U_{\text{acc}}^h, L_{\text{fair}}^h)$ represents the best-case accuracy-fairness trade-off achieved by $h$. If the confidence region lies entirely in the sub-optimal region, we can confidently conclude that $h$ achieves a suboptimal trade-off.

(b) Here, $(L_{\text{acc}}^h, U_{\text{fair}}^h)$ represents the worst-case accuracy-fairness trade-off achieved by $h$. If the confidence region lies entirely in the 'unlikely' region, we can conclude that $h$ achieves a better trade-off than YOTO.

Figure 13: Confidence region $C^\alpha(h)$ for the accuracy-fairness trade-off achieved by $h$.

*Then,* $\mathbb{P}((\text{acc}(h'), \Phi_{\text{fair}}(h')) \in [0, U_{\text{acc}}^h] \times [L_{\text{fair}}^h, 1]) \geq 1 - \alpha.$

Lemma E.1 shows that $C^\alpha(h') = [0, U_{\text{acc}}^h] \times [L_{\text{fair}}^h, 1]$ forms the $1 - \alpha$ confidence region for the accuracy-fairness trade-off $(\text{acc}(h'), \Phi_{\text{fair}}(h'))$. We illustrate this in Figure 13a. If this confidence region lies entirely above the permissible region (i.e. in the pink region in Figure 13), we can confidently conclude that the model $h'$ achieves suboptimal accuracy-fairness trade-off. From Figure 13a it can be seen that this will occur if $(U_{\text{acc}}^h, L_{\text{fair}}^h)$ lies above the permissible region.

Intuitively, $(U_{\text{acc}}^h, L_{\text{fair}}^h)$ can be seen as an optimistic best-case accuracy-fairness trade-off achieved by the model $h'$, since at this point in the confidence region $C^\alpha(h')$ the accuracy is maximised and fairness violation is minimised. Therefore if this best-case trade-off lies above the permissible region, then this intuitively indicates that the worst-case optimal trade-off $\tau_{\text{fair}}^*$ is still better than the best-case trade-off achieved by the model $h'$, leading us to the confident conclusion that $h'$ achieves suboptimal accuracy-fairness trade-off.

**Baseline's worst-case accuracy-fairness trade-off** Conversely, if the confidence region $C^\alpha(h')$ lies entirely below the confidence intervals $\Gamma_{\text{fair}}^\alpha$ (i.e. in the blue region in Figure 13), then using union bounds we can confidently conclude that with probability $1 - 2\alpha$ we have that the model $h'$ achieves a better trade-off than the YOTO model $h_\theta(\cdot, \lambda)$. Formally, this means that

$$\tau^{\text{YOTO}}(\text{acc}(h')) \geq \Phi_{\text{fair}}(h'),$$

where

$$\tau^{\text{YOTO}}(\delta) := \min_{\lambda \in \Lambda} \Phi_{\text{fair}}(h_\theta(\cdot, \lambda)) \quad \text{subject to} \quad \text{acc}(h_\theta(\cdot, \lambda)) \geq \delta.$$

This would indicate that the YOTO model does not achieve the optimal trade-off and can be used to further calibrate the $\Delta(h_\lambda)$ values when constructing the lower confidence interval using Proposition 3.3. Again, this is different from simply checking if the empirical trade-off achieved by the model $(\widehat{\text{acc}}(h'), \widehat{\Phi_{\text{fair}}}(h'))$ lies in the unlikely-to-be-achieved trade-off region (blue region in Figure 13) as it also accounts for the finite-sampling uncertainty in the tradeoff achieved by model $h'$.

Next, we show to construct such confidence region $C^\alpha(h')$ using an approach analogous to the one outlined above:

**Lemma E.2.** *For a classifier $h' \in \mathcal{H}$, let $L_{\text{acc}}^h, U_{\text{fair}}^h$ be the lower and upper CIs on $\text{acc}(h'), \Phi_{\text{fair}}(h')$ respectively,*

$$\mathbb{P}(\text{acc}(h') \geq L_{\text{acc}}^h) \geq 1 - \alpha/2, \quad \text{and} \quad \mathbb{P}(\Phi_{\text{fair}}(h') \leq U_{\text{fair}}^h) \geq 1 - \alpha/2.$$

*Then,* $\mathbb{P}((\text{acc}(h'), \Phi_{\text{fair}}(h')) \in [L_{\text{acc}}^h, 1] \times [0, U_{\text{fair}}^h]) \geq 1 - \alpha.$

Table 5: The table shows the proportion of models whose best-case trade-off $(U_{\text{acc}}^h, L_{\text{fair}}^h)$ lies in the suboptimal region, models whose worst-case trade-off $(L_{\text{acc}}^h, L_{\text{fair}}^h)$ lies in the unlikely region and models whose confidence regions $C^\alpha(h)$ overlaps with the permissible trade-off region. The table reports average across all experiments.

| BASELINE | UNLIKELY | PERMISSIBLE | SUB-OPTIMAL |
|---|---|---|---|
| KDE-FAIR | $0.0 \pm 0.0$ | $1.0 \pm 0.0$ | $0.0 \pm 0.0$ |
| RTO | $0.0 \pm 0.0$ | $0.77 \pm 0.05$ | $0.23 \pm 0.04$ |
| ADVERSARY | $0.0 \pm 0.0$ | $0.79 \pm 0.1$ | $0.2 \pm 0.08$ |
| LINEAR | $0.08 \pm 0.05$ | $0.81 \pm 0.06$ | $0.11 \pm 0.05$ |
| LOGSIG | $0.05 \pm 0.02$ | $0.71 \pm 0.09$ | $0.25 \pm 0.1$ |
| REDUCTIONS | $0.0 \pm 0.0$ | $0.82 \pm 0.09$ | $0.18 \pm 0.08$ |
| SEPARATE | $0.03 \pm 0.02$ | $0.95 \pm 0.08$ | $0.02 \pm 0.03$ |

Lemma E.2 shows that $C^\alpha(h') = [L_{\text{acc}}^h, 1] \times [0, U_{\text{fair}}^h]$ forms the $1 - \alpha$ confidence region for the accuracy-fairness trade-off $(\text{acc}(h'), \Phi_{\text{fair}}(h'))$. If this confidence region lies entirely below the permissible region (i.e. in the blue region in Figure 13), we can confidently conclude that the model $h'$ achieves a better accuracy-fairness trade-off than the YOTO model. From Figure 13b it can be seen that this will occur if $(L_{\text{acc}}^h, U_{\text{fair}}^h)$ lies below the permissible region.

Intuitively, $(L_{\text{acc}}^h, U_{\text{fair}}^h)$ can be seen as a conservative worst-case accuracy-fairness trade-off achieved by the model $h'$. Therefore if this worst-case trade-off lies below the permissible region, then this intuitively indicates that the best-case YOTO trade-off $\tau^{\text{YOTO}}$ is still worse than the worst-case trade-off achieved by the model $h'$, leading us to the confident conclusion that $h'$ achieves a better accuracy-fairness trade-off than the YOTO model. This can subsequently be used to calibrate the suboptimality gap for the YOTO model, denoted by $\Delta(h_\lambda)$ in Proposition 3.3.

## E.1 Experimental results

Here, we present the results with the empirical baseline trade-offs replaced by best or worst-case trade-offs as appropriate. More specifically, in Figure 14,

- if the empirical trade-off for a baseline $(\widehat{\text{acc}}(h'), \widehat{\Phi_{\text{fair}}}(h'))$ lies in the suboptimal region (as in Figure 13a), then we plot the best-case trade-off $(U_{\text{acc}}^h, L_{\text{fair}}^h)$ for the baseline,

- if the empirical trade-off for a baseline $(\widehat{\text{acc}}(h'), \widehat{\Phi_{\text{fair}}}(h'))$ lies in the unlikely-to-be-achievable trade-off region (as in Figure 13b), then we plot the worst-case trade-off $(L_{\text{acc}}^h, U_{\text{fair}}^h)$ for the baseline,

- if the empirical trade-off for a baseline $(\widehat{\text{acc}}(h'), \widehat{\Phi_{\text{fair}}}(h'))$ lies in the permissible region, then we simply plot the empirical trade-off $(\widehat{\text{acc}}(h'), \widehat{\Phi_{\text{fair}}}(h'))$.

Therefore, in Figure 14 if a baseline's best-case trade-off lies above the permissible trade-off region, then we can confidently conclude that the baseline achieves suboptimal accuracy-fairness trade-off with probability $1 - 2\alpha = 90\%$. Similarly, a baseline's worst-case trade-off lying below the permissible trade-off region would suggest that the YOTO trade-off achieves a suboptimal trade-off and that the value of $\Delta(h_\theta)$ needs to be adjusted accordingly.

Table 5 shows the proportion of best-case trade-offs which lie above the permissible trade-off region and the proportion of worst-case trade-offs which lie below the permissible region. Firstly, the table shows that the proportion of worst-case trade-offs which lie in the 'unlikely' region is negligible, empirically confirming that our confidence intervals on optimal trade-off $\tau_{\text{fair}}^*$ are indeed valid. Secondly, we can see that there are a considerable proportion of baselines whose best-case trade-off lies above the permissible region, highlighting that our methodology remains effective in flagging suboptimalities in SOTA baselines even when we account for the possible uncertainty in baseline trade-offs. This shows that our methodology yields CIs which are not only reliable but also informative.

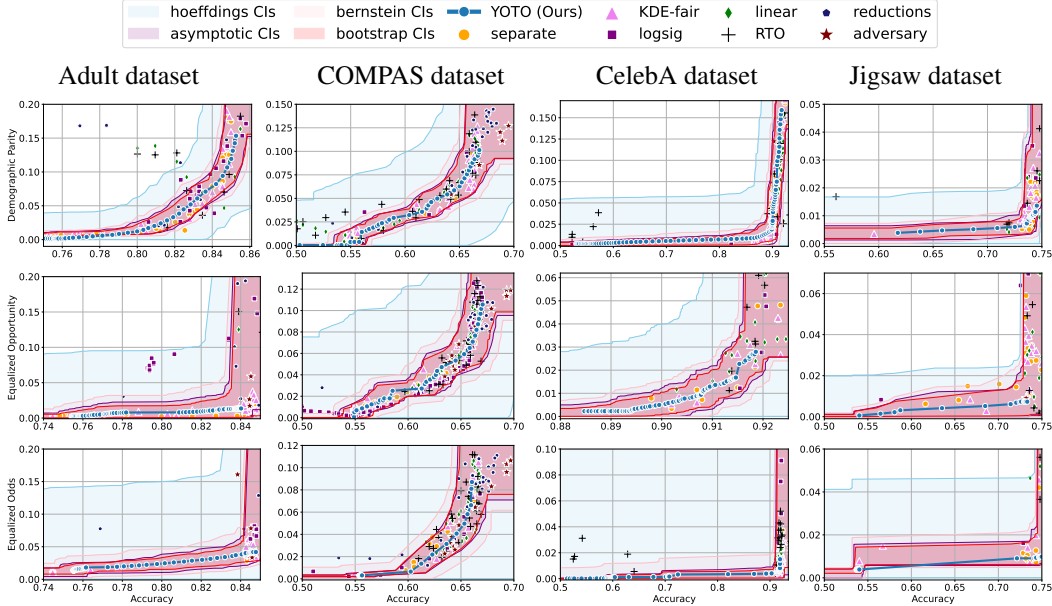

Figure 14: Results with the empirical baseline trade-offs replaced by best-case and worst-case trade-offs for baselines lying above and below the permissible region respectively. Here, $\mathcal{D}_{cal}$ is 10% data split and $\alpha = 0.05$.

## F  Experimental details and additional results

In this section, we provide greater details regarding our experimental setup and models used. We first begin by defining the Equalized Odds metric which has been used in our experiments, along with DP and EOP.

**Equalized Odds (EO):** EO condition states that, both the true positive rates and false positive rates for all sensitive groups are equal, i.e. $\mathbb{P}(h(X) = 1 \mid A = a, Y = y) = \mathbb{P}(h(X) = 1 \mid Y = y)$ for any $a \in \mathcal{A}$ and $y \in \{0, 1\}$. The absolute EO violation is defined as:

$$\Phi_{\mathrm{EO}}(h) := 1/2 \left| \mathbb{P}(h(X) = 1 \mid A = 1, Y = 1) - \mathbb{P}(h(X) = 1 \mid A = 0, Y = 1) \right|$$
$$+ 1/2 \left| \mathbb{P}(h(X) = 1 \mid A = 1, Y = 0) - \mathbb{P}(h(X) = 1 \mid A = 0, Y = 0) \right|.$$

Next, we provide additional details regarding the YOTO model.

### F.1  Practical details regarding YOTO model

As described in Section 3, we consider optimising regularized losses of the form

$$\mathcal{L}_\lambda(\theta) = \mathbb{E}[l_{\mathrm{CE}}(h_\theta(X), Y)] + \lambda \, \mathcal{L}_{\mathrm{fair}}(h_\theta).$$

When training YOTO models, instead of fixing $\lambda$, we sample the parameter $\lambda$ from a distribution $P_\lambda$. As a result, during training the model observes many different values of $\lambda$ and learns to optimise the loss $\mathcal{L}_\lambda$ for all of them simultaneously. At inference time, the model can be conditioned on a chosen parameter value $\lambda'$ and recovers the model trained to optimise $\mathcal{L}_{\lambda'}$. The loss being minimised can thus be expressed as follows:

$$\underset{h_\theta : \mathcal{X} \times \Lambda \to \mathbb{R}}{\arg\min} \; \mathbb{E}_{\lambda \sim P_\lambda} \left[ \mathbb{E}[l_{\mathrm{CE}}(h_\theta(X, \lambda), Y)] + \lambda \, \mathcal{L}_{\mathrm{fair}}(h_\theta(\cdot, \lambda)) \right].$$

The fairness losses $\mathcal{L}_{\mathrm{fair}}$ considered for the YOTO model are:

DP:  $\mathcal{L}_{\mathrm{fair}}(h_\theta(\cdot, \lambda)) = |\mathbb{E}[\sigma(h_\theta(X, \lambda)) \mid A = 1] - \mathbb{E}[\sigma(h_\theta(X, \lambda)) \mid A = 0]|$

EOP:  $\mathcal{L}_{\mathrm{fair}}(h_\theta(\cdot, \lambda)) = |\mathbb{E}[\sigma(h_\theta(X, \lambda)) \mid A = 1, Y = 1] - \mathbb{E}[\sigma(h_\theta(X, \lambda)) \mid A = 0, Y = 1]|$

EO:  $\mathcal{L}_{\mathrm{fair}}(h_\theta(\cdot, \lambda)) = |\mathbb{E}[\sigma(h_\theta(X, \lambda)) \mid A = 1, Y = 1] - \mathbb{E}[\sigma(h_\theta(X, \lambda)) \mid A = 0, Y = 1]|$
$+ |\mathbb{E}[\sigma(h_\theta(X, \lambda)) \mid A = 1, Y = 0] - \mathbb{E}[\sigma(h_\theta(X, \lambda)) \mid A = 0, Y = 0]|.$

Here, $\sigma(x) := 1/(1 + e^{-x})$ denotes the sigmoid function.

In our experiments, we sample a new $\lambda$ for every batch. Moreover, we use the log-uniform distribution as per [15] as the sampling distribution $P_\lambda$, where the uniform distribution is $U[10^{-6}, 10]$. To condition the network on $\lambda$ parameters, we follow in the footsteps of [15] to use Feature-wise Linear Modulation (FiLM) [34]. For completeness, we include the description of the architecture next.

Initially, we determine which network layers should be conditioned, which can encompass all layers or just a subset. For each chosen layer, we condition it based on the weight parameters $\lambda$. Given a layer that yields a feature map $f$ with dimensions $W \times H \times C$, where $W$ and $H$ denote the spatial dimensions and $C$ stands for the channels, we introduce the parameter vector $\lambda$ to two distinct multi-layer perceptrons (MLPs), denoted as $M_\sigma$ and $M_\mu$. These MLPs produce two vectors, $\sigma$ and $\mu$, each having a dimensionality of $C$. The feature map is then transformed by multiplying it channel-wise with $\sigma$ and subsequently adding $\mu$. The resultant transformed feature map $f'$ is given by:

$$f'_{ijk} = \sigma_k f_{ijk} + \mu_k \quad \text{where} \quad \sigma = M_\sigma(\lambda) \quad \text{and} \quad \mu = M_\mu(\lambda).$$

Next, we provide exact architectures we used for each dataset in our experiments.

### F.1.1 YOTO Architectures

**Adult and COMPAS dataset** Here, we use a simple logistic regression as the main model, with only the scalar logit outputs of the logistic regression being conditioned using FiLM. The MLPs $M_\mu, M_\sigma$ both have two hidden layers, each of size 4, and ReLU activations. We train the model for a maximum of 1000 epochs, with early stopping based on validation losses. Training these simple models takes roughly 5 minutes on a Tesla-V100-SXM2-32GB GPU.

**CelebA dataset** For the CelebA dataset, our architecture is a convolutional neural network (ConvNet) integrated with the FiLM (Feature-wise Linear Modulation) mechanism. The network starts with two convolutional layers: the first layer has 32 filters with a kernel size of $3 \times 3$, and the second layer has 64 filters, also with a $3 \times 3$ kernel. Both convolutional layers employ a stride of 1 and are followed by a max-pooling layer that reduces each dimension by half.

The feature maps from the convolutional layers are flattened and passed through a series of fully connected (MLP) layers. Specifically, the first layer maps the features to 64 dimensions, and the subsequent layers maintain this size until the final layer, which outputs a scalar value. The activation function used in these layers is ReLU.

To condition the network on the $\lambda$ parameter using FiLM, we design two multi-layer perceptrons (MLPs), $M_\mu$ and $M_\sigma$. Both MLPs take the $\lambda$ parameter as input and have 4 hidden layers. Each of these hidden layers is of size 256. These MLPs produce the modulation parameters $\mu$ and $\sigma$, which are used to perform feature-wise linear modulation on the outputs of the main MLP layers. The final output of the network is passed through a sigmoid activation function to produce the model's prediction. We train the model for a maximum of 1000 epochs, with early stopping based on validation losses. Training this model takes roughly 1.5 hours on a Tesla-V100-SXM2-32GB GPU.

**Jigsaw dataset** For the Jigsaw dataset, we employ a neural network model built upon the BERT architecture [13] integrated with the Feature-wise Linear Modulation (FiLM) mechanism. We utilize the representation corresponding to the [CLS] token, which carries aggregate information about the entire sequence. To condition the BERT's output on the $\lambda$ parameter using FiLM, we design two linear layers, which map the $\lambda$ parameter to modulation parameters $\gamma$ and $\beta$, both of dimension equal to BERT's hidden size of 768. These modulation parameters are then used to perform feature-wise linear modulation on the [CLS] representation. The modulated representation is passed through a classification head, which consists of a linear layer mapping from BERT's hidden size (768) to a scalar output. In terms of training details, our model is trained for a maximum of 10 epochs, with early stopping based on validation losses. Training this model takes roughly 6 hours on a Tesla-V100-SXM2-32GB GPU.

### F.2 Datasets

We used four real-world datasets for our experiments.

**Adult dataset** The Adult income dataset [7] includes employment data for 48,842 individuals where the task is to predict whether an individual earns more than \$50k per year and includes demographic

attributes such as age, race and gender. In our experiments, we consider gender as the sensitive attribute.

**COMPAS dataset** The COMPAS recidivism data comprises collected by ProPublica [5], includes information for 6172 defendants from Broward County, Florida. This information comprises 52 features including defendants' criminal history and demographic attributes, and the task is to predict recidivism for defendants. The sensitive attribute in this dataset is the defendants' race where $A = 1$ represents 'African American' and $A = 0$ corresponds to all other races.

**CelebA dataset** The CelebA dataset [28] consists of 202,599 celebrity images annotated with 40 attribute labels. In our task, the objective is to predict whether an individual in the image is smiling. The dataset comprises features in the form of image pixels and additional attributes such as hairstyle, eyeglasses, and more. The sensitive attribute for our experiments is gender.

**Jigsaw Toxicity Classification dataset** The Jigsaw Toxicity Classification dataset [20] contains online comments from various platforms, aimed at identifying and mitigating toxic behavior online. The task is to predict whether a given comment is toxic or not. The dataset includes features such as the text of the comment and certain metadata such as the gender or race to which each comment relates. In our experiments, the sensitive attribute is the gender to which the comment refers, and we only filter the comments which refer to exactly one of 'male' or 'female' gender. This leaves us with 107,106 distinct comments.

### F.3 Baselines

The baselines considered in our experiments include:

- **Regularization based approaches [8]:** These methods seek to minimise fairness loss using regularized losses as shown in Section 3. We consider different regularization terms $\mathcal{L}_{\text{fair}}(h_\theta)$ as smooth relaxations of the fairness violation $\Phi_{\text{fair}}$ as proposed in the literature [8]. To make this concrete, when the fairness violation under consideration is DP, we consider

$$\mathcal{L}_{\text{fair}}(h_\theta) = \mathbb{E}[g(h_\theta(X)) \mid A = 1] - \mathbb{E}[g(h_\theta(X)) \mid A = 0],$$

with $g(x) = x$ denoted as 'linear' in our results and $g(x) = \log \sigma(x)$ where $\sigma(x) := 1/(1 + e^{-x})$, denoted as 'logsig' in our results. In addition to these methods, we also consider separately trained models with the same regularization term as the YOTO models, i.e.,

DP:    $\mathcal{L}_{\text{fair}}(h_\theta) = |\mathbb{E}[\sigma(h_\theta(X)) \mid A = 1] - \mathbb{E}[\sigma(h_\theta(X)) \mid A = 0]|$

EOP:    $\mathcal{L}_{\text{fair}}(h_\theta) = |\mathbb{E}[\sigma(h_\theta(X)) \mid A = 1, Y = 1] - \mathbb{E}[\sigma(h_\theta(X)) \mid A = 0, Y = 1]|$

EO:    $\mathcal{L}_{\text{fair}}(h_\theta) = |\mathbb{E}[\sigma(h_\theta(X)) \mid A = 1, Y = 1] - \mathbb{E}[\sigma(h_\theta(X)) \mid A = 0, Y = 1]|$
       $+ |\mathbb{E}[\sigma(h_\theta(X)) \mid A = 1, Y = 0] - \mathbb{E}[\sigma(h_\theta(X)) \mid A = 0, Y = 0]|.$

Here, $\sigma(x) := 1/(1 + e^{-x})$ denotes the sigmoid function. We denote these models as 'separate' in our experimental results as they are the separately trained counterparts to the YOTO model. For each relaxation, we train models for a range of $\lambda$ values uniformly chosen in $[0, 10]$ interval.

- **Reductions Approach [1]:** This method transforms the fairness problem into a sequence of cost-sensitive classification problems. Like the regularization approaches this requires multiple models to be trained. Here, to try and reproduce the trade-off curves, we train the reductions approach with a range of different fairness constraints uniformly in $[0, 1]$.

- **KDE-fair [12]:** This method employs a kernel density estimation trick to quantify fairness measures, capturing the degree of irrelevancy of prediction outputs to sensitive attributes. These quantified fairness measures are expressed as differentiable functions with respect to classifier model parameters, allowing the use of gradient descent to solve optimization problems that respect fairness constraints. We focus on binary classification and well-known definitions of group fairness: Demographic Parity (DP), Equalized Odds (EO) and Equalized Opportunity (EOP).

- **Randomized Threshold Optimizer (RTO) [2]:** This scalable post-processing algorithm debiases trained models, including deep neural networks, and is proven to be near-optimal by bounding its excess Bayes risk. RTO optimizes the trade-off curve by applying adjustments

to the trained model's predictions to meet fairness constraints. In our experiments, we first train a standard classifier which is referred to as the base classifier. Next, we use the RTO algorithm to apply adjustments to the trained model's prediction to meet fairness constraints. To reproduce the trade-off curves, we apply post-hoc adjustments to the base classifier for a range of fairness violation constraints.

- **Adversarial Approaches [45]:** These methods utilize an adversarial training paradigm where an additional model, termed the adversary, is introduced during training. The primary objective of this adversary is to predict the sensitive attribute $A$ using the predictions $h_\theta(X)$ generated by the main classifier $h_\theta$. The training process involves an adversarial game between the primary classifier and the adversary, striving to achieve equilibrium. This adversarial dynamic ensures that the primary classifier's predictions are difficult to use for determining the sensitive attribute $A$, thereby minimizing unfair biases associated with $A$. Specifically, for DP constraints, the adversary takes the logit outputs of the classifier as the input and predicts $A$. In contrast for EO and EOP constraints, the adversary also takes the true label $Y$ as the input. For EOP constraint, the adversary is only trained on data with $Y = 1$.

The training times for each baseline on the different datasets have been listed in Table 6.

Table 6: Approximate training times per model for different baselines across various datasets.

|  | Adult | COMPAS | CelebA | Jigsaw |
|---|---|---|---|---|
| adversary | 3 min | 3 min | 90 min | 310 min |
| linear/logsig/separate | 4 min | 3 min | 80 min | 310 min |
| KDE-fair | 2 min | 2 min | 60 min | 280 min |
| reductions | 3 min | 2 min | 70 min | 290 min |
| YOTO | 4 min | 3 min | 90 min | 320 min |
| RTO (base classifier training) | 3 min | 3 min | 70 min | 310 min |
| RTO (Post-hoc optimisation per fairness constraint) | 1 min | 1 min | 10 min | 30 min |

### F.4 Additional results

In Figures 15-26 we include additional results for all datasets and fairness violations with an increasing number of calibration data $\mathcal{D}_{\text{cal}}$. It can be seen that as the number of calibration data increases, the CIs constructed become increasingly tighter. However, the asymptotic, Bernstein and bootstrap CIs are informative even when the calibration data is as little as 500 for COMPAS data (Figures 18-20) and 1000 for all other datasets. These results show that the larger the calibration data $\mathcal{D}_{\text{cal}}$, the tighter the constructed CIs are likely to be. However, even in cases where the calibration dataset is relatively small, we obtain informative CIs in most cases.

Tables 7 - 18 show the proportion of the baselines which lie in the three trade-off regions, 'Unlikely', 'Permissible' and 'Sub-optimal' for the Bernstein's CIs, across the different datasets and fairness metrics. Overall, it can be seen that our CIs are reliable since the proportion of trade-offs which lie below our CIs is small. On the other hand, the tables show that our CIs can detect a significant number of sub-optimalities in our baselines, showing that our intervals are informative.

### F.5 Additional details for Figure 1

For Figure 1, we consider a calibration data $\mathcal{D}_{\text{cal}}$ of size 2000. For the optimal model, we trained a YOTO model on the COMPAS dataset with the architecture given in F.1.1. We train the model for a maximum of 1000 epochs, with early stopping based on validation losses. Once trained, we obtained the red and black curves by first splitting $\mathcal{D}_{\text{cal}}$ randomly into two subsets, and evaluating the trade-offs on each split separately. On the other hand, for the sub-optimal model, we use another YOTO model with the same architecture but stop the training after 20 epochs. This results in the trained model achieving a sub-optimal trade-off as shown in the figure. We evaluated this trade-off curve on the entire calibration data.

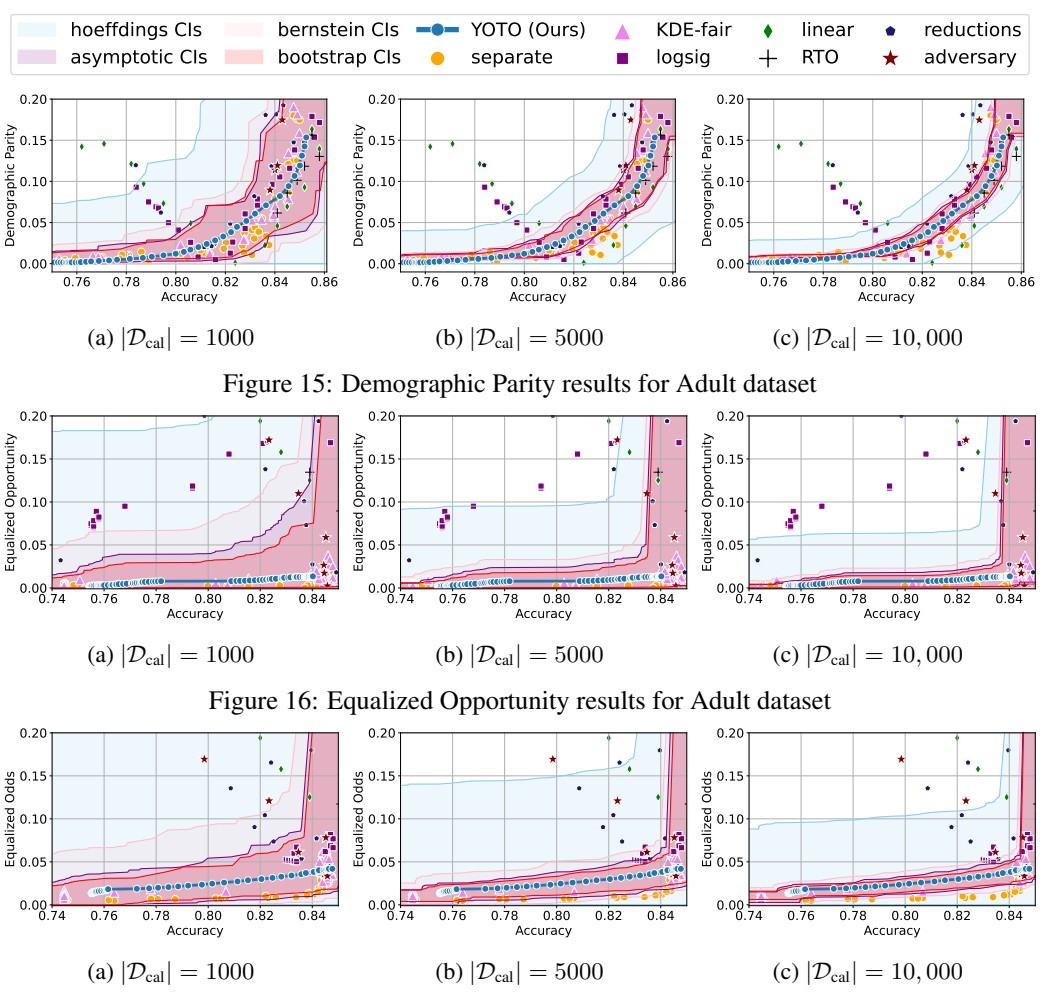

Figure 15: Demographic Parity results for Adult dataset

Figure 16: Equalized Opportunity results for Adult dataset

Figure 17: Equalized Odds results for Adult dataset

Table 7: Proportion of empirical trade-offs for each baseline which lie in the three trade-off regions, for the COMPAS dataset and DP (using Bernstein's CIs). Here $\mathcal{D}_{\text{cal}}$ is a 10% dataset split.

|  | UNLIKELY | PLAUSIBLE | SUB-OPTIMAL |
|---|---|---|---|
| KDE-FAIR | 0.0 | 1.0 | 0.0 |
| RTO | 0.0 | 0.67 | 0.33 |
| ADVERSARY | 0.15 | 0.85 | 0.0 |
| LINEAR | 0.0 | 0.82 | 0.18 |
| LOGSIG | 0.05 | 0.2 | 0.75 |
| REDUCTIONS | 0.0 | 0.83 | 0.17 |
| SEPARATE | 0.0 | 1.0 | 0.0 |

## F.6 Experimental results with FACT Frontiers

**MA FACT and MS FACT yield conservative Pareto Frontiers.** In Figure 27 we include the results for the Adult dataset along with the model-specific (MS) and model-agnostic (MA) FACT Pareto frontiers, obtained using the methodology in [23]. It can be seen that both Pareto frontiers, MS FACT and MA FACT are overly conservative, with the MA FACT being completely non-informative on the Adult dataset. The model-agnostic FACT trade-off considers the best-case trade-off for any given dataset and does not depend on a specific model class. As a result, there is no guarantee that this Pareto frontier is achievable. This is evident from the fact that the MA FACT yields a non-informative Pareto frontier on the Adult dataset in Figure 27, which is non-achievable in practice.

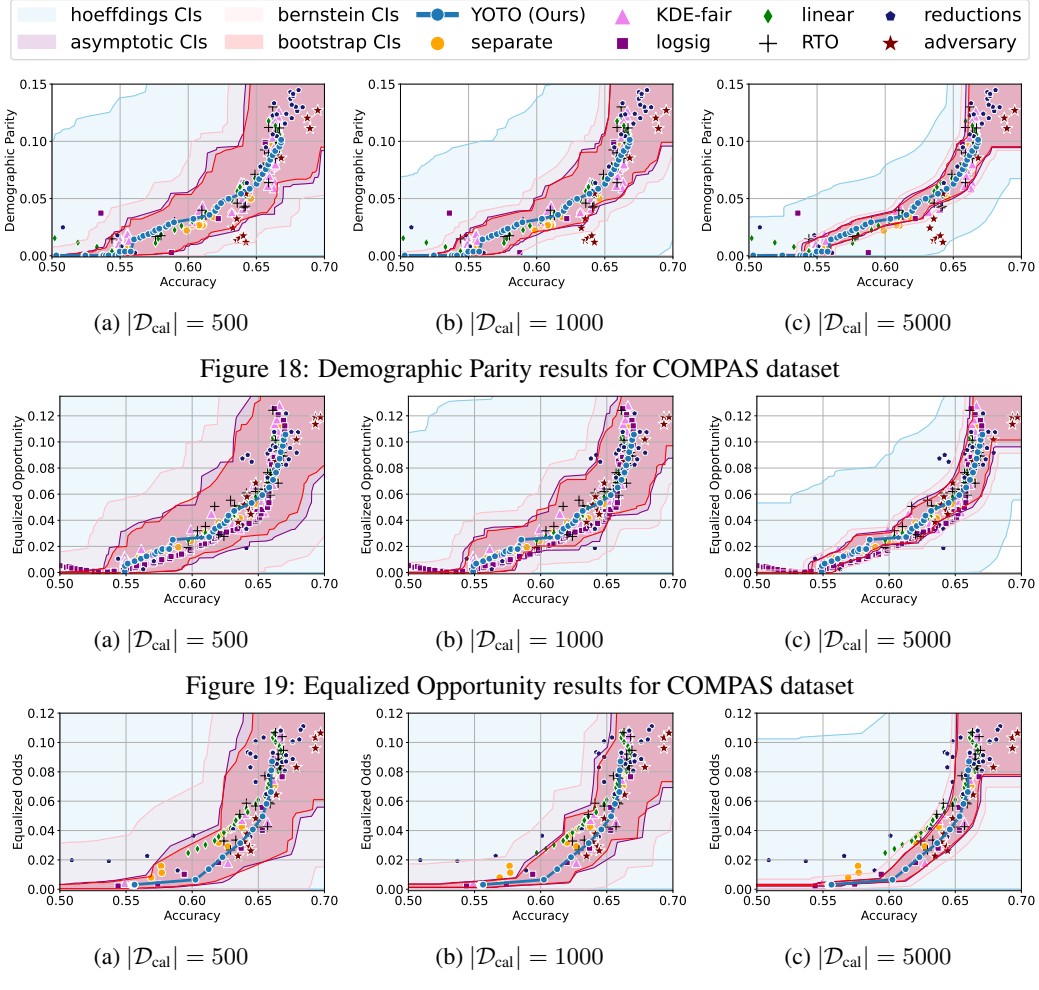

Figure 18: Demographic Parity results for COMPAS dataset

Figure 19: Equalized Opportunity results for COMPAS dataset

Figure 20: Equalized Odds results for COMPAS dataset

Table 8: Proportion of empirical trade-offs for each baseline which lie in the three trade-off regions, for the COMPAS dataset and EOP (using Bernstein's CIs). Here $\mathcal{D}_{\text{cal}}$ is a 10% dataset split.

|  | UNLIKELY | PLAUSIBLE | SUB-OPTIMAL |
| --- | --- | --- | --- |
| KDE-FAIR | 0.0 | 0.99 | 0.01 |
| RTO | 0.0 | 0.96 | 0.04 |
| ADVERSARY | 0.0 | 1.0 | 0.0 |
| LINEAR | 0.0 | 1.0 | 0.0 |
| LOGSIG | 0.01 | 0.36 | 0.62 |
| REDUCTIONS | 0.03 | 0.77 | 0.2 |
| SEPARATE | 0.0 | 1.0 | 0.0 |

The model-specific FACT trade-off curve is comparatively more realistic as it is derived using a pre-trained model. However, it is important to note that the trade-off for this pre-trained model may not lie on the obtained MS FACT frontier. In fact, MS FACT may still not be achievable by any model in the given model class, as it does not provide any guarantee of being achievable either. This can be seen from Figure 27 which shows that the MS-FACT frontier is not achieved by any of the SOTA baselines under consideration. In these experiments, we used a logistic regression classifier as a pre-trained model for obtaining the MS FACT frontier.

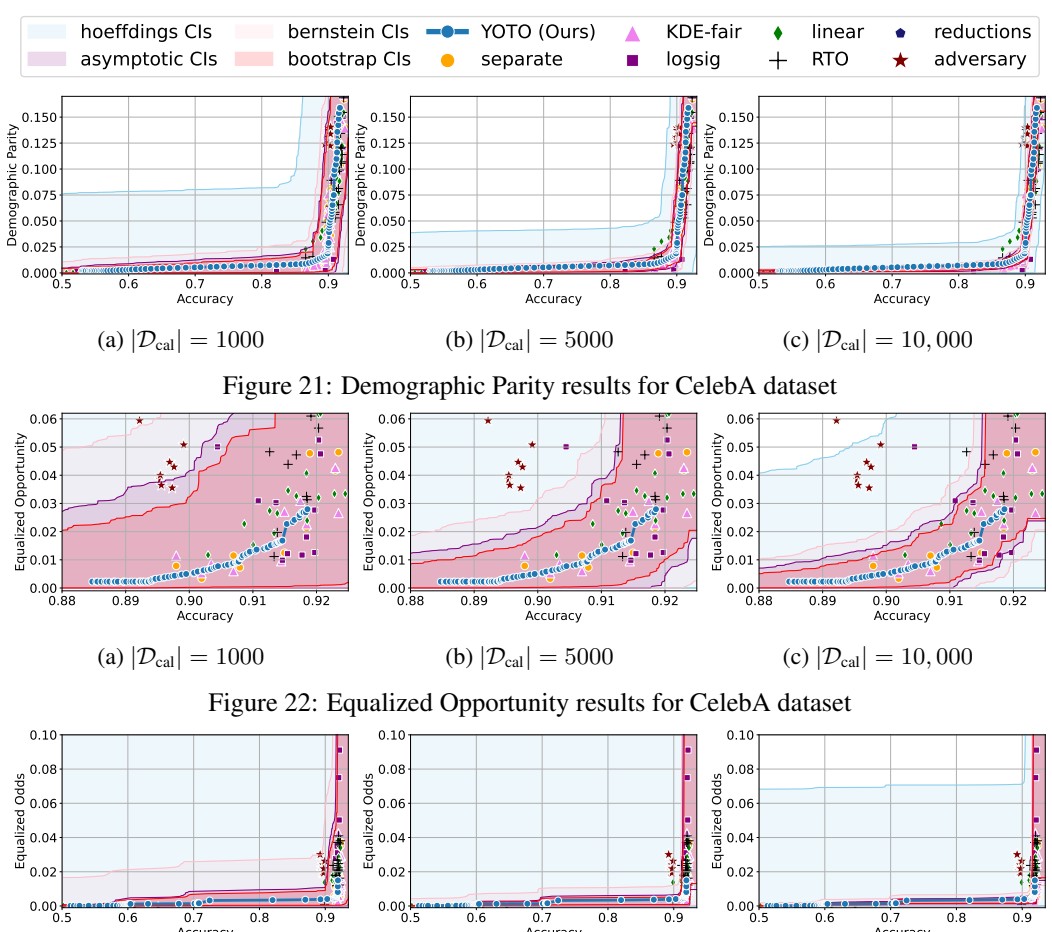

(a) $|\mathcal{D}_{\text{cal}}| = 1000$       (b) $|\mathcal{D}_{\text{cal}}| = 5000$       (c) $|\mathcal{D}_{\text{cal}}| = 10,000$

Figure 21: Demographic Parity results for CelebA dataset

(a) $|\mathcal{D}_{\text{cal}}| = 1000$       (b) $|\mathcal{D}_{\text{cal}}| = 5000$       (c) $|\mathcal{D}_{\text{cal}}| = 10,000$

Figure 22: Equalized Opportunity results for CelebA dataset

(a) $|\mathcal{D}_{\text{cal}}| = 1000$       (b) $|\mathcal{D}_{\text{cal}}| = 5000$       (c) $|\mathcal{D}_{\text{cal}}| = 10,000$

Figure 23: Equalized Odds results for CelebA dataset

Table 9: Proportion of empirical trade-offs for each baseline which lie in the three trade-off regions, for the COMPAS dataset and EO (using Bernstein's CIs). Here $\mathcal{D}_{\text{cal}}$ is a 10% dataset split.

|  | UNLIKELY | PLAUSIBLE | SUB-OPTIMAL |
|---|---|---|---|
| KDE-FAIR | 0.0 | 1.0 | 0.0 |
| RTO | 0.0 | 0.89 | 0.11 |
| ADVERSARY | 0.0 | 1.0 | 0.0 |
| LINEAR | 0.0 | 0.76 | 0.24 |
| LOGSIG | 0.0 | 0.45 | 0.55 |
| REDUCTIONS | 0.0 | 0.7 | 0.3 |
| SEPARATE | 0.1 | 0.87 | 0.03 |

## F.7  Synthetic data experiments

In real-world settings, the ground truth trade-off curve $\tau^*_{\text{fair}}$ remains intractable because we only have access to a finite dataset. In this section, we consider a synthetic data setting, where the ground truth trade-off curve can be obtained, to verify that the YOTO trade-off curves are consistent with the ground truth and that the confidence intervals obtained using our methodology contain $\tau^*_{\text{fair}}$.

**Dataset** Here, we consider a setup with $\mathcal{X} = \mathbb{R}$, $\mathcal{A} = \{0, 1\}$ and $\mathcal{Y} = \{0, 1\}$. Specifically, $A \sim \text{Bern}(0.5)$ and we define the conditional distributions $X \mid A = a$ as:

$$X \mid A = a \sim \mathcal{N}(a, 0.2^2)$$

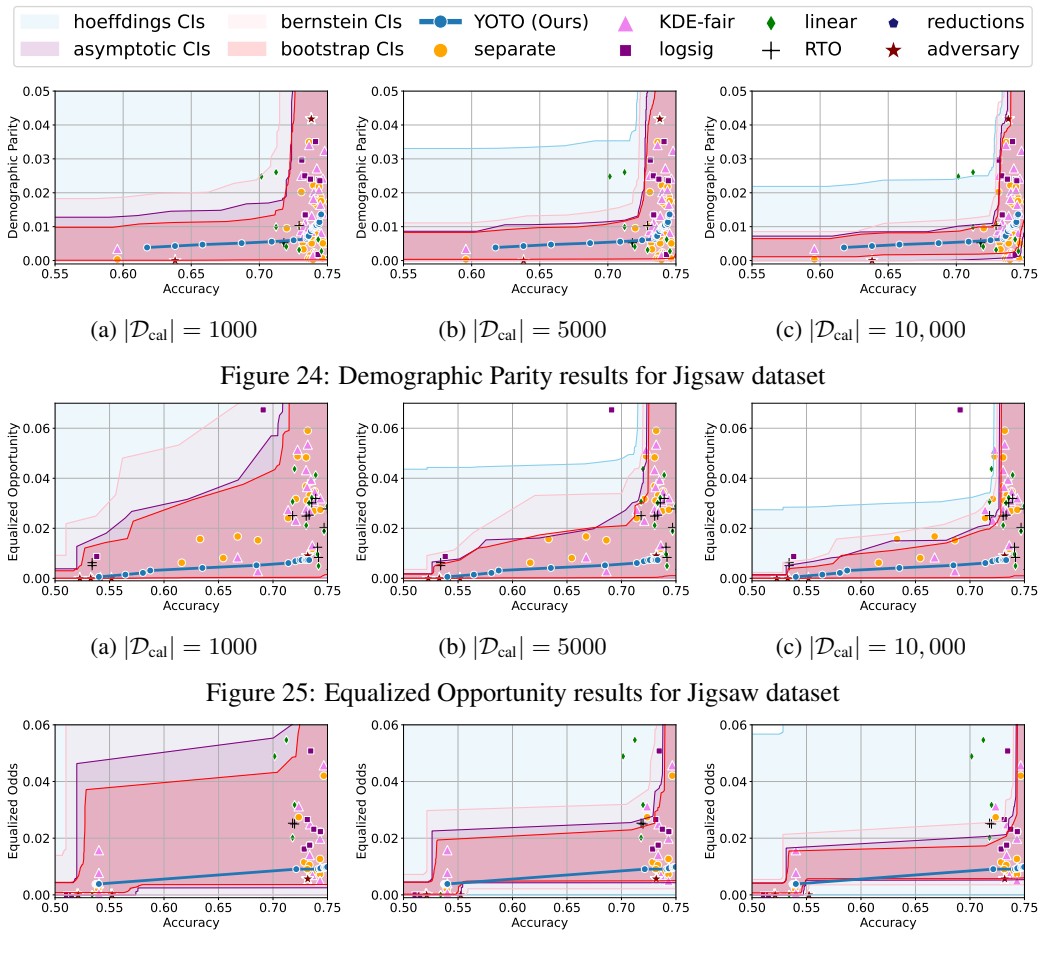

Figure 24: Demographic Parity results for Jigsaw dataset

Figure 25: Equalized Opportunity results for Jigsaw dataset

Figure 26: Equalized Odds results for Jigsaw dataset

Table 10: Proportion of empirical trade-offs for each baseline which lie in the three trade-off regions, for the Adult dataset and DP (using Bernstein's CIs). Here $\mathcal{D}_{\text{cal}}$ is a 10% dataset split.

|  | UNLIKELY | PLAUSIBLE | SUB-OPTIMAL |
|---|---|---|---|
| KDE-FAIR | 0.0 | 1.0 | 0.0 |
| RTO | 0.0 | 0.33 | 0.67 |
| ADVERSARY | 0.0 | 0.0 | 1.0 |
| LINEAR | 0.13 | 0.47 | 0.4 |
| LOGSIG | 0.0 | 0.27 | 0.73 |
| REDUCTIONS | 0.0 | 0.13 | 0.87 |
| SEPARATE | 0.03 | 0.95 | 0.02 |

Moreover, we define the labels $Y$ as follows:

$$Y = Z \, \mathbb{1}(X > 0.5) + (1 - Z) \, \mathbb{1}(X \leq 0.5),$$

where $Z \sim \text{Bern}(0.9)$ and $Z \perp\!\!\!\perp X$. Here, $Z$ introduces some 'noise' to the labels $Y$ and means that perfect accuracy is not achievable by linear classifiers. If perfect accuracy was achievable, the optimal values for Equalized Odds and Equalized Opportunity would be 0 (and would be achieved by the perfect classifier), therefore our use of 'noisy' labels $Y$ ensures that the ground truth trade-off curves will be non-trivial.

**YOTO model training** Using the data generating mechanism, we generate 5000 training datapoints, which we use to train the YOTO model. The YOTO model for this dataset comprises of a simple

Table 11: Proportion of empirical trade-offs for each baseline which lie in the three trade-off regions, for the Adult dataset and EOP (using Bernstein's CIs). Here $\mathcal{D}_{\text{cal}}$ is a 10% dataset split.

|  | UNLIKELY | PLAUSIBLE | SUB-OPTIMAL |
|---|---|---|---|
| KDE-FAIR | 0.07 | 0.89 | 0.04 |
| RTO | 0.0 | 0.5 | 0.5 |
| ADVERSARY | 0.0 | 0.36 | 0.64 |
| LINEAR | 0.0 | 0.5 | 0.5 |
| LOGSIG | 0.0 | 0.09 | 0.91 |
| REDUCTIONS | 0.0 | 0.17 | 0.83 |
| SEPARATE | 0.02 | 0.95 | 0.03 |

Table 12: Proportion of empirical trade-offs for each baseline which lie in the three trade-off regions, for the Adult dataset and EO (using Bernstein's CIs). Here $\mathcal{D}_{\text{cal}}$ is a 10% dataset split.

|  | UNLIKELY | PLAUSIBLE | SUB-OPTIMAL |
|---|---|---|---|
| KDE-FAIR | 0.0 | 1.0 | 0.0 |
| RTO | 0.0 | 0.4 | 0.6 |
| ADVERSARY | 0.0 | 0.36 | 0.64 |
| LINEAR | 0.0 | 0.4 | 0.6 |
| LOGSIG | 0.0 | 0.65 | 0.35 |
| REDUCTIONS | 0.0 | 0.1 | 0.9 |
| SEPARATE | 0.0 | 1.0 | 0.0 |

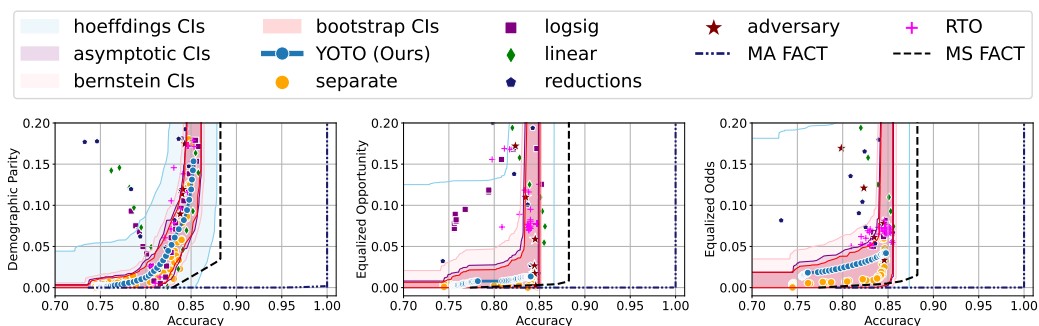

Figure 27: Results on the Adult dataset with FACT frontiers (pre-sensitivity analysis, i.e. $\Delta(h_\lambda) = 0$). Here, $|\mathcal{D}_{\text{cal}}| = 2000$ and $\alpha = 0.05$.

logistic regression as the main model, with only the scalar logit outputs of the logistic regression being conditioned using FiLM. The MLPs $M_\mu, M_\sigma$ both have two hidden layers, each of size 4, and ReLU activations. We train the model for a maximum of 1000 epochs, with early stopping based on validation losses. Training these simple model only requires one CPU and takes roughly 2 minutes.

**Ground truth trade-off curve** To obtain the ground truth trade-off curve $\tau^*_{\text{fair}}$, we consider the family of classifiers

$$h_c(X) = \mathbb{1}(X > c)$$

for $c \in \mathbb{R}$. Next, we calculate the trade-offs achieved by this model family for a fine grid of $c$ values between -3 and 3, using a dataset of size 500,000 obtained using the data-generating mechanism described above. The large dataset size ensures that the finite sample errors in accuracy and fairness violation values are negligible. This allows us to reliably plot the trade-off curve $\tau^*_{\text{fair}}$.

### F.7.1 Results

Figure 28 shows the results for the synthetic data setup for three different fairness violations, obtained using a calibration dataset $\mathcal{D}_{\text{cal}}$ of size 2000. It can be seen that for each fairness violation considered, the YOTO trade-off curve aligns very well with the ground-truth trade-off curve $\tau^*_{\text{fair}}$. Additionally, we also consider four different confidence intervals obtained using our methodology, and Figure 28

Table 13: Proportion of empirical trade-offs for each baseline which lie in the three trade-off regions, for the CelebA dataset and DP (using Bernstein's CIs). Here $\mathcal{D}_{\text{cal}}$ is a 10% dataset split.

| | UNLIKELY | PLAUSIBLE | SUB-OPTIMAL |
|---|---|---|---|
| KDE-FAIR | 0.0 | 1.0 | 0.0 |
| RTO | 0.0 | 0.56 | 0.44 |
| ADVERSARY | 0.0 | 0.18 | 0.82 |
| LINEAR | 0.0 | 0.88 | 0.12 |
| LOGSIG | 0.05 | 0.95 | 0.0 |
| SEPARATE | 0.0 | 1.0 | 0.0 |

Table 14: Proportion of empirical trade-offs for each baseline which lie in the three trade-off regions, for the CelebA dataset and EOP (using Bernstein's CIs). Here $\mathcal{D}_{\text{cal}}$ is a 10% dataset split.

| | UNLIKELY | PLAUSIBLE | SUB-OPTIMAL |
|---|---|---|---|
| KDE-FAIR | 0.0 | 0.91 | 0.09 |
| RTO | 0.0 | 0.45 | 0.55 |
| ADVERSARY | 0.0 | 0.27 | 0.73 |
| LINEAR | 0.15 | 0.82 | 0.03 |
| LOGSIG | 0.0 | 0.73 | 0.27 |
| SEPARATE | 0.0 | 1.0 | 0.0 |

shows that each of the four confidence intervals considered contain the ground-truth trade-off curve. This empirically verifies the validity of our confidence intervals in this synthetic setting.

Additionally, in Figure 29 we plot how the worst-case values for $\Delta(h)$ change (relative to the optimal trade-off $\tau_{\text{fair}}^*$) as the number of training data $\mathcal{D}_{\text{tr}}$ increases. Here, we use $h_\lambda$ as a short-hand notation for the YOTO model $h_\theta(\cdot, \lambda)$ and the quantity on the $y$-axis, $\max_{\lambda \in \Lambda} \frac{\Delta(h_\lambda)}{\tau_{\text{fair}}^*(\text{acc}(h_\lambda))}$, can be intuitively thought of as the worst-case error percentage between YOTO trade-off and the optimal trade-off. The figure empirically verifies our result in Theorem 3.4 by showing that as the number of training data increases, the error term declines across all fairness metrics.

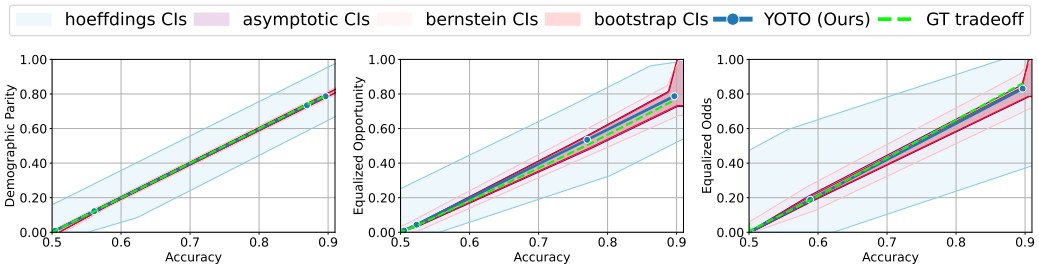

Figure 28: Results for the synthetic dataset with ground truth trade-off curves $\tau_{\text{fair}}^*$.

Table 15: Proportion of empirical trade-offs for each baseline which lie in the three trade-off regions, for the CelebA dataset and EO (using Bernstein's CIs). Here $\mathcal{D}_{\mathrm{cal}}$ is a 10% dataset split.

|            | UNLIKELY | PLAUSIBLE | SUB-OPTIMAL |
|------------|----------|-----------|-------------|
| KDE-FAIR   | 0.0      | 1.0       | 0.0         |
| RTO        | 0.0      | 0.71      | 0.29        |
| ADVERSARY  | 0.0      | 0.27      | 0.73        |
| LINEAR     | 0.0      | 0.97      | 0.03        |
| LOGSIG     | 0.0      | 0.73      | 0.27        |
| SEPARATE   | 0.0      | 1.0       | 0.0         |

Table 16: Proportion of empirical trade-offs for each baseline which lie in the three trade-off regions, for the Jigsaw dataset and DP (using Bernstein's CIs). Here $\mathcal{D}_{\mathrm{cal}}$ is a 10% dataset split.

|            | UNLIKELY | PLAUSIBLE | SUB-OPTIMAL |
|------------|----------|-----------|-------------|
| KDE-FAIR   | 0.0      | 0.92      | 0.08        |
| RTO        | 0.0      | 0.5       | 0.5         |
| ADVERSARY  | 0.0      | 0.09      | 0.91        |
| LINEAR     | 0.0      | 0.76      | 0.24        |
| LOGSIG     | 0.0      | 0.4       | 0.6         |
| SEPARATE   | 0.0      | 0.88      | 0.12        |

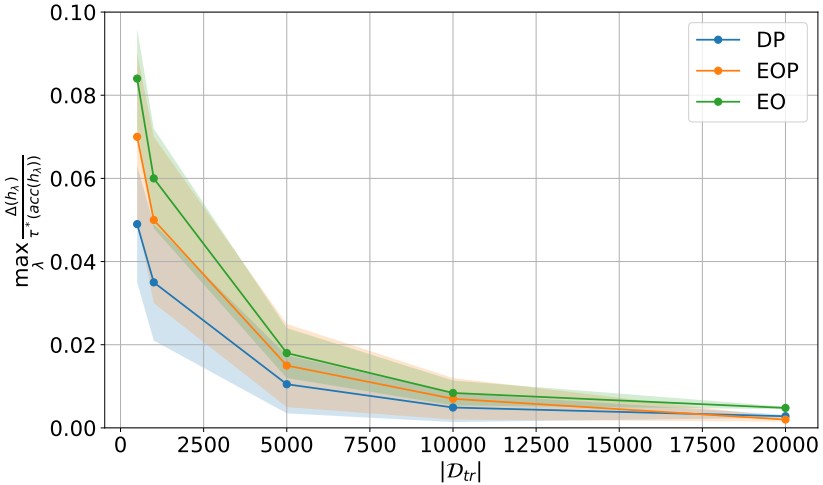

Figure 29: Plot showing how $\Delta(h)$ decreases (relative to the ground truth trade-off value $\tau^*_{\mathrm{fair}}(\mathrm{acc}(h))$) as the training data size $|\mathcal{D}_{\mathrm{tr}}|$ increases. Here, we plot the worst (i.e. largest) value of $\Delta(h_\lambda)/\tau^*_{\mathrm{fair}}(\mathrm{acc}(h_\lambda))$ achieved by our YOTO model over a grid of $\lambda$ values in $[0, 5]$.

Table 17: Proportion of empirical trade-offs for each baseline which lie in the three trade-off regions, for the Jigsaw dataset and EOP (using Bernstein's CIs). Here $\mathcal{D}_{\mathrm{cal}}$ is a 10% dataset split.

|            | UNLIKELY | PLAUSIBLE | SUB-OPTIMAL |
|------------|----------|-----------|-------------|
| KDE-FAIR   | 0.0      | 0.83      | 0.17        |
| RTO        | 0.0      | 0.62      | 0.38        |
| ADVERSARY  | 0.06     | 0.94      | 0.0         |
| LINEAR     | 0.14     | 0.59      | 0.27        |
| LOGSIG     | 0.0      | 0.29      | 0.71        |
| SEPARATE   | 0.0      | 0.47      | 0.53        |

Table 18: Proportion of empirical trade-offs for each baseline which lie in the three trade-off regions, for the Jigsaw dataset and EO (using Bernstein's CIs). Here $\mathcal{D}_{\mathrm{cal}}$ is a 10% dataset split.

| | UNLIKELY | PLAUSIBLE | SUB-OPTIMAL |
|---|---|---|---|
| KDE-FAIR | 0.0 | 0.75 | 0.25 |
| RTO | 0.0 | 0.57 | 0.43 |
| ADVERSARY | 0.0 | 1.0 | 0.0 |
| LINEAR | 0.0 | 0.73 | 0.27 |
| LOGSIG | 0.0 | 0.4 | 0.6 |
| SEPARATE | 0.0 | 0.87 | 0.13 |

