# OpenReview forum: "Achievable Fairness on Your Data With Utility Guarantees"
_NeurIPS.cc/2024/Conference — NeurIPS 2024 poster_

### Official Review · Reviewer_Vp4R · 2024-07-06

**Soundness:** 3
**Presentation:** 3
**Contribution:** 3
**Rating:** 7
**Confidence:** 4

**Summary:**

This paper addresses a significant limitation in the fairness literature, which is the use of uniform fairness requirements across diverse datasets. It proposes the YOTO framework to approximate the fairness-accuracy trade-off and reduce computational costs while existing methods typically require multiple models for training.

**Strengths:**

- Originality: This paper discusses the problems associated with using uniform fairness metrics across diverse datasets and provides a framework for choosing fairness guidelines based on the characteristics of individual datasets. While this issue has been addressed in the fairness literature, this paper proposes a new framework to tackle it. Unlike previous work that recovers a single point on the trade-off curve corresponding to pre-specified fairness constraints during the training procedure, this method estimates the entire trade-off curve.

- Quality and Clarity: The paper is well-constructed and is easy-to-follow. The motivation and research question are clearly stated. The theoretical analysis is solid. The experiment is conducted across three different practical applications (tabular data, image, and text), which is rigorous.

- Significance: This paper proposes a novel framework to approximate the accuracy-fairness trade-off curve, which is an important and efficient evaluation paradigm for providing fairness guidelines in practice. The method has a practical advantage, allowing decision-makers to observe the minimum fairness violation that a particular dataset could have as the model accuracy increases.

**Weaknesses:**

- The content in lines 121-126 that illustrates the suboptimal problem, as shown in Figure 1, is highly similar to the content in lines 61-66. This content should be summarized to make it more concise.
- Since this work is based on YOTO, YOTO should be introduced in the Preliminary section for better illustration.
- Section 3.1 should also be introduced in the Preliminary section, as it is the basic in-processing framework for fairness.
- There is a lack of dataset and experiment setup descriptions, including data size, sensitive attributes chosen, held-out set size, and train-test split ratio.

**Questions:**

In Lemma 3.1, $\delta$ specifies the relationship between the confidence interval and the data size of the held-out set. Is there any requirement for the size of the held-out calibration dataset to guarantee an effective measure for the confidence intervals?

**Limitations:**

The authors have adequately addressed the limitations. Since this work is mainly about fairness, I do not see any negative societal impact.

---

> ### Author Rebuttal · Authors · 2024-08-06
>
> We sincerely appreciate the reviewer's recognition of the quality and originality of our work. We clarify the questions raised below.
>
> > The content in lines 121-126 that illustrates the suboptimal problem, as shown in Figure 1, is highly similar to the content in lines 61-66. This content should be summarized to make it more concise [...]
> Since this work is based on YOTO, YOTO should be introduced in the Preliminary section for better illustration [...]
> Section 3.1 should also be introduced in the Preliminary section, as it is the basic in-processing framework for fairness [...]
>
> We thank the reviewer for their suggestions regarding the structure of our write-up and will make the proposed modifications to the final version of our paper.
>
> > There is a lack of dataset and experiment setup descriptions, including data size, sensitive attributes chosen, held-out set size, and train-test split ratio.
>
> Due to space constraints, we have included comprehensive details about our experimental setup in Appendix F of our paper. This includes dataset details (including dataset sizes), sensitive attributes chosen, model architectures used for both, our YOTO model and the baselines, and training details (such as the number of epochs, compute resources used, methodology of early stopping and fairness losses). For completeness, we will also include the key details in the final version of our main text.
>
> > In Lemma 3.1, $\delta$ specifies the relationship between the confidence interval and the data size of the held-out set. Is there any requirement for the size of the held-out calibration dataset to guarantee an effective measure for the confidence intervals?
>
> In general, Hoeffding's inequality provides finite-sample guarantees which remain valid regardless of the size of the calibration dataset. However, if the calibration dataset size is low, the CIs obtained might be more conservative (and hence less informative).
> The width of the CIs decreases as the calibration dataset size increases, thereby leading to more informative confidence intervals. In fact, it can be seen from Lemma 3.1 that the width of the CIs on $acc(h_\lambda)$ is $\mathcal{O}(1/\sqrt{|D_{cal}|})$.
>
> To investigate how the calibration dataset size impacts the informativeness of our CIs in practice, we have conducted comprehensive ablation studies across different datasets in Appendix F.4. Our empirical results show that while the informativeness of the CIs for a given calibration dataset size depends on the fairness metric and dataset under consideration, we observe that in many cases the CIs obtained are informative when the size of calibration dataset is at least 1000.
>
> We hope the above addressed the concerns raised by the reviewer and that the reviewer will consider increasing their score.

---

> > ### Comment · Reviewer_Vp4R · 2024-08-08
> >
> > I am satisfied with the authors’ replies, and they have addressed my concerns and questions. After checking the replies the authors provided to other reviewers, I am happy to raise my score.

---

### Official Review · Reviewer_NxJb · 2024-07-10

**Soundness:** 3
**Presentation:** 3
**Contribution:** 3
**Rating:** 6
**Confidence:** 4

**Summary:**

Considering inherent accuracy-fairness trade-off in real-world scenarios with data imbalance and bias, imposing strict fairness constraint could be impractical. To address this, the paper introduces an efficient method to contextualize and accurately estimating fairness-accuracy trade-off curve for each dataset. The proposed YOTO ease redundant model training to realize the trade-off curves. Also, the paper presents a method to quantify the confidence of the estimate.

**Strengths:**

- Clarity of the writing: The paper is easy to follow and well-motivated.
- Novel Approach: The paper introduces a novel method to approximate the fairness-accuracy trade-off curve efficiently, which is a significant improvement over traditional methods requiring multiple model trainings.
- Computational Efficiency: By leveraging the YOTO framework, the proposed method reduces the computational burden, making it feasible for large datasets and complex models.
- Practical Utility: The framework provides a practical tool for auditing model fairness, offering a principled way to select data-specific fairness requirements and assess compliance.

**Weaknesses:**

- Data Dependency: The methodology requires separate datasets for training and calibration, which may not be feasible in situations with limited data.
- Assumptions for Statistical Guarantees: The statistical guarantees rely on assumptions that may not hold in all practical scenarios, potentially limiting the generalizability of the results.
- Objective Formulation Dependency: The trend (scattered points) of YOTO and corresponding CIs are specifically designed for relaxed fairness metric rather than explicit fairness notion, which may make the boundary inaccurate. In other words, trade-off curve and CIs can be sensitive to fairness loss formulation.

**Questions:**

- Can I understand the trade-off curve estimated by proposed method similar to Pareto Frontier with CIs inherent in dataset when model design is specified? If so, I think it would be worth mentioning related thresholding post-processing methods [1,2].
- I think we can easily interpret points reside in upper-left are suboptimal in Figure 3. However, how can we interpret the ones reside in lower-right region? Does this mean the proposed trade-off estimate and corresponding CI is not accurate?



[1] Kim, Joon Sik, Jiahao Chen, and Ameet Talwalkar. "FACT: A diagnostic for group fairness trade-offs." International Conference on Machine Learning. PMLR, 2020.

[2] Jang, Taeuk, Pengyi Shi, and Xiaoqian Wang. "Group-aware threshold adaptation for fair classification." Proceedings of the AAAI Conference on Artificial Intelligence. Vol. 36. No. 6. 2022.

**Limitations:**

Please see the questions and weaknesses above.

---

> ### Author Rebuttal · Authors · 2024-08-06
>
> We are grateful to the reviewer for acknowledging the novelty and practical utility of our approach and the clarity of our writing. We respond to the questions raised below.
>
> > Assumptions for Statistical Guarantees: The statistical guarantees rely on assumptions that may not hold in all practical scenarios, potentially limiting the generalizability of the results.
>
> We would like to clarify that while Theorem 3.4 relies on regularity assumptions, our statistical guarantee in Proposition 3.2 does not require any additional assumptions beyond the exchangeability of data.
> In specific, Proposition 3.2 provides the 'worst-case' optimal trade-off, accounting for finite-sample uncertainty and the upper CIs obtained remain valid even if the YOTO classifier $h_\lambda$ is not trained well (and hence achieves sub-optimal accuracy-fairness trade-off), although in such cases the CI may be conservative.
> This means that any trade-off which lies above the upper CIs is guaranteed to be suboptimal with probability $1-\alpha$ without any additional assumptions.
>
> > Objective Formulation Dependency: [...] trade-off curves and CIs can be sensitive to fairness loss formulation
>
> While the trade-offs and CIs obtained may be sensitive to the fairness loss formulation, it is important to emphasise that our coverage guarantees in Propositions 3.2 and 3.3 are independent of the fairness loss formulations.
> Specifically, our upper CIs obtained in Proposition 3.2 remain valid regardless of the fairness loss used.
> Similarly, for the lower CIs, our sensitivity analysis should adjust the lower CIs to account for any sub-optimalities in the estimated trade-off curve arising from our choice of fairness loss in practice.
>
> This is also evident from our experimental results in Section 5 where we consider baselines with various fairness loss formulations, and show that the obtained trade-offs are overall consistent with our CIs.
>
> > Can I understand the trade-off curve estimated by proposed method similar to Pareto Frontier with CIs inherent in dataset when model design is specified? If so, I think it would be worth mentioning related thresholding post-processing methods [1,2].
>
> We thank the reviewer for their question. It is important to clarify the distinct goals of our work and those in [1,2]. Our focus in this work is on reliably approximating the optimal accuracy-fairness trade-offs achievable within a specific model class, $\mathcal{H}$, using the finite data available. To achieve this, we construct CIs with probabilistic guarantees of the form $\mathbb{P}(\tau^*_{fair}(\Psi) \in \Gamma^\alpha) \geq 1-\alpha$.
>
> This is in contrast to [1, 2] which provide upper bounds on the best achievable accuracy-fairness trade-off when infinite data is available. While these works offer valuable theoretical insights, there is no guarantee regarding whether this upper bound is achievable by the model class under consideration, given available data. In fact, we consider the FACT Pareto frontiers proposed in [1] in Appendix F.6, where we show empirically that these Pareto frontiers are highly conservative and are not achieved by any of the SOTA baselines we considered. We will further highlight these distinctions in our updated manuscript.
>
> > I think we can easily interpret points reside in upper-left are suboptimal in Figure 3. However, how can we interpret the ones reside in lower-right region? Does this mean the proposed trade-off estimate and corresponding CI is not accurate?
>
> Our confidence intervals (CIs) quantify the range of most likely trade-offs achievable for a given finite dataset and model class.
> In particular, our CIs only offer coverage of at least $1-\alpha$.
> This means that if a baseline's trade-off falls below the CIs (i.e. in the blue region in Figure 1) then this would suggest that the model achieves an exceptionally good accuracy-fairness trade-off, although in practice this will rarely occur (i.e. with a probability of at most $\alpha$).
> This is also consistent with our experimental results, where we consider
> $\alpha = 0.05$. Our results in Table 1 verify that less than 5\% of the SOTA baseline trade-offs fall below our lower CIs, as promised by Proposition 3.3.
>
> We hope that the above has addressed the questions raised by the reviewer adequately and that the reviewer will consider raising their score.

---

### Official Review · Reviewer_3zFr · 2024-07-14

**Soundness:** 3
**Presentation:** 3
**Contribution:** 3
**Rating:** 7
**Confidence:** 4

**Summary:**

This paper proposes a computationally efficient method to estimate the accuracy-fairness trade-off curve with the statistical guarantee given the dataset and model class. Specifically, it first adopts an existing method, You-Only-Train-Once (YOTO), to get the trade-off curve and then proposes a systematic way to obtain the confidence intervals for fairness and accuracy.

**Strengths:**

1. The research problem on the fairness-accuracy tradeoff is fundamental and practical. There is rarely literature that touches on this problem.
2. Considering the finite sample error, the confidence interval is essential for tradeoff investigation. This paper proposes a systematic way to investigate this problem.
3. The paper is well-written and easy to follow.

**Weaknesses:**

1. The research focus, the fairness-accuracy tradeoff, is computationally expensive since the auditing process usually involves multiple model training. Even though the author adopts a computation-efficient YOTO framework to tackle this issue. However, the main contribution on the efficiency side is mainly from existing work. I suggest the authors highlight the values of statistical guarantee. For example, given the statistical guarantee, can the authors provide a more comprehensive evaluation or any insights for model training? Otherwise, the audience may have a "so what" question in mind.
2. I understand the general clue for deriving confidence intervals for the fairness and accuracy side, but I am still confused about how the authors estimate confidence intervals in practice, especially for sensitivity $\Delta(h_{\lambda})$. (1) How do you estimate or justify $\Delta(h_{\lambda})$ without know Pareto optimality? It seems to be counter-intuitive. (2) Do you consider Pareto optimality only for specific model classes? I am curious about "When YOTO satisfies Pareto optimality" (Line 693). If the main conclusion only holds under specific model classes, it would be better to highlight such condition. (3) I suspect the analysis is rigorous. For example, for Hoeffding’s inequality in Line 215, $\tilde{acc}$ should be the expectation of $acc$. However, $\tilde{acc}$ is the accuracy of $h_{\lambda}$ on $D_{cal}$, which is a realization (not mean) of ACC variable.
3. I notice that there is a similar literature at https://openreview.net/forum?id=dSbbZwCTQI, which also trains once with flexible tradeoffs. The main idea of YODO is to ``line'' in the weight space that connects the accuracy-optimum and fairness-optimum points using a single model, which is different from YOTO manner. It would be better to highlight the difference and why the authors chose YOTO instead of YODO.

**Questions:**

Please see Weakness.

**Limitations:**

The method requires an in-distribution calibration dataset and only applies to in-distribution tests.

---

> ### Author Rebuttal · Authors · 2024-08-06
>
> We sincerely thank the reviewer for their thoughtful review and for highlighting the strengths of our work, including its motivation and presentation. Below we address some of the questions raised.
> > [...] given the statistical guarantee, can the authors provide a more comprehensive evaluation or any insights for model training? [...]
>
> Our confidence intervals (CIs) are designed to equip practitioners with an auditing tool to assess fairness trade-offs for different model classes and datasets.
>
> If for a classifier $h_0$, the accuracy-fairness trade-off lies above our proposed upper CIs (i.e. in the pink region in Figure 1), then with probability $1-\alpha$ the classifier $h_0$ achieves a sub-optimal trade-off (i.e., there exists another classifier $h' \in \mathcal{H}$ with $acc(h') \geq acc(h_0)$ and $\Phi_{fair}(h') \leq \Phi_{fair}(h_0)$).
> In this case, one should explore alternative fairness methodologies and model architectures until the model achieves a trade-off in the "permissible trade-off region" (green area in Figure 1). In this way, our CIs can also guide algorithm development, as researchers can use the permissible region of trade-offs identified by our CIs as a target during the design of improved algorithms.
>
> It is worth emphasising that unlike prior works which only consider the empirical accuracy-fairness trade-offs, our CI-based methodology is the first to avoid false conclusions arising from finite-sampling errors, thereby offering more reliable insights in realistic data-limited scenarios, as we demonstrate empirically in Figure 1.
>
> > How do you estimate or justify $\Delta(h_\lambda)$ without known Pareto optimality?
>
> First, we recall that $\Delta(h_\lambda)$ (defined in Figure 2a) intuitively denotes the gap between the trade-off achieved by YOTO and the optimal achievable trade-off $\tau^*_{fair}$, and is an unknown quantity in general. Therefore, for practical purposes, we use sensitivity analysis to posit 'plausible' values for this quantity.
> The high-level idea behind our procedure is to calibrate the value of $\Delta(h_\lambda)$ using some additional models trained separately using standard methods with varying fairness constraints.
>
> **Sensitivity analysis**
> Explicitly, our sensitivity analysis approach is as follows: First, aside from our YOTO model, our sensitivity analysis procedure uses $k$ additional models $\mathcal{M} :=$ { $h_1, ..., h_k$ } trained separately using standard regularized loss (in Eqn. (2) of our paper).
>
> Let $\mathcal{M_0} \subseteq \mathcal{M}$ be the models which achieve a better empirical trade-off than our YOTO model (see Figure 2b in our main text). We choose $\Delta(h_\lambda)$ for our YOTO model to be the maximum gap between the empirical trade-offs of these models in $\mathcal{M}_0$ and the YOTO model. In practice, this shifts our lower CIs downward until all the trade-offs for models in $\mathcal{M}$ lie above our lower CIs.
>
> **Intuition**
> If we assume that training models separately for each fairness constraint recovers the optimal trade-off curve, then we could in principle obtain this optimal curve by training many different models, although at a significant computational cost. Our approach instead offers a compromise between the optimality of trade-offs and computational cost, as we instead train a few models with fairness regularization parameters $\lambda$ sampled uniformly and use these to posit plausible values for $\Delta(h_\lambda)$. (In our experiments, we found that using 2 separately trained models for sensitivity analysis was sufficient.)
>
> > Do you consider Pareto optimality only for specific model classes?
>
> Our methodology remains valid for any model class $\mathcal{H}$ which can be optimised using gradient-based methods. In this setting, the models in $\mathcal{H}$ can be trained using a YOTO-style architecture and hence our methodology remains applicable. While we mention this in Section 2.1 (lines 108-109), we will highlight this further in the updated manuscript.
>
> > [...] for Hoeffding’s inequality in Line 215, $\tilde{acc}$ should be the expectation of $acc$ [...]
>
> We wish to clarify that  $\widetilde{acc(h_\lambda)}$ is indeed a random variable. Formally,
>     $$
> \widetilde{acc(h_\lambda)} := \sum_{(X_i, Y_i) \in D_{cal}} \frac{1(h_\lambda(X_i) = Y_i)}{|D_{cal}|}.
>     $$
>     Then, Hoeffding's inequality shows that with probability at least $1-\alpha$ we have that
>     $$
>     |\widetilde{acc(h_\lambda)} - \mathbb{E}[\widetilde{acc(h_\lambda)}] | \leq \sqrt{\log{(2/\alpha)}/(2|D_{cal}|)}.
>     $$
>     Then, Lemma 3.1 follows from the fact that $acc(h_\lambda) = \mathbb{E}[\widetilde{acc(h_\lambda)}]$.
>
> We will clarify this further in the final version.
>
> > [...] It would be better to highlight the difference and why the authors chose YOTO instead of YODO
>
> Firstly, we thank the reviewer for pointing us to this manuscript. The YODO architecture comprises two sets of parameters, each corresponding to accuracy and fairness optimisation respectively.
> This means that, for a given size of the main model, the YODO architecture will require the optimisation of twice as many parameters as the standard model.
>
> In contrast, the YOTO architecture does not split the weights but instead uses the FiLM layers to dynamically adapt the hidden layers of the models depending on the fairness regularization parameter $\lambda$.
> Consequently, YOTO only adds a relatively small number of additional parameters as it only involves two additional MLPs which can be significantly smaller than the main model. (See Appendix F.1 for more details on model architectures.) As a result, training YOTO can be computationally cheaper, especially in cases where the main model is very large.
> For completeness, we will include a comprehensive discussion of this comparison in the final version of our paper.
>
> We hope that we were able to address all the reviewer's questions and hope that the reviewer would consider increasing their score.

---

> > ### Comment · Reviewer_3zFr · 2024-08-12
> > **Official Comment by Reviewer 3zFr**
> >
> > Thanks for the detailed response. I am satisfied with the response and will raise my score.

---

### Decision · Program_Chairs · 2024-09-25

**Decision:**

Accept (poster)

**Comment:**

The authors introduce a new method to estimate the fairness-accuracy tradeoff (and the uncertainty over this tradeoff) in a single training run. This method involves adapting the previously proposed YOTO framework to compute suitable confidence intervals over the dataset at hand. The reviewers were excited by the novelty of the proposed approach and the potential for more sample efficient fair learning.